# $\alpha$-DPO: Robust Preference Alignment for Diffusion Model via $\alpha$ Divergence

**Yang Li**[1,2]  **Songlin Yang**[3]  **Wei Wang**[1*]  **Xiaoxuan Han**[1,2]  **& Dong Jing**[1]
[1]New Laboratory of Pattern Recognition, MAIS, CASIA
[2]School of Artificial Intelligence, UCAS  [3]The Hong Kong University of Science and Technology

## ABSTRACT

Diffusion models have demonstrated remarkable success in high-fidelity image generation, yet aligning them with human preferences remains challenging. Direct Preference Optimization (DPO) offers a promising framework, but its effectiveness is critically hindered by noisy data arising from mislabeled preference pairs and individual preference pairs. We theoretically show that existing DPO objectives are equivalent to minimizing the Forward Kullback–Leibler (KL) divergence, whose mass-covering nature makes it intrinsically sensitive to such noise. To address this limitation, we propose $\alpha$-DPO, which reformulates preference alignment through the lens of $\alpha$-divergence. This formulation promotes mode-seeking behavior and bounds the influence of outliers, thereby enhancing robustness. Furthermore, we introduce a dynamic scheduling mechanism that adaptively adjusts $\alpha$ according to the observed preference distribution, providing data-aware noise tolerance during training. Extensive experiments on synthetic and real-world datasets validate that $\alpha$-DPO consistently outperforms existing baselines, achieving superior robustness and preference alignment. The code and project page are available at `https://github.com/yangli-lab/Diffusion_alpha-DPO_ICLR2026/`.

## 1 INTRODUCTION

Diffusion models (Labs, 2024; Esser et al., 2024; Podell et al., 2023) have recently emerged as powerful generative frameworks capable of producing high-fidelity and photorealistic images. However, beyond visual quality, aligning these models with human preferences is crucial for practical deployment, since users often care about semantic relevance, stylistic choices, and aesthetic qualities that are not fully captured by likelihood-based training. A common paradigm to address this alignment problem is reinforcement learning from human feedback (RLHF) (Black et al., 2023; Fan et al., 2023; Zhu et al., 2025; Yang et al., 2024), which leverages preference annotations to train a reward model and subsequently optimize the generator via reinforcement learning. While effective, RLHF introduces additional complexity and instability due to the need for explicit reward modeling and policy optimization. Direct Preference Optimization (DPO) (Wallace et al., 2024) offers a more streamlined alternative: instead of constructing a separate reward function, DPO directly optimizes the generative model to increase the likelihood of human-preferred samples over less preferred ones, thus achieving preference alignment in a simpler and more stable manner.

However, the effectiveness of DPO paradigms critically hinges on the quality of their preference data. In practice, such datasets inevitably contain noisy preference pairs from two primary sources: (i) mislabeled preference pairs, where annotation errors introduce signals that contradict real preferences; and (ii) individual preference pairs, arising from annotator subjectivity, where either assignment of winner and loser can be considered valid ("Also OK" labels). Although these noisy pairs originate differently, they both alter the intended preference label. From this perspective, we unify them as label-flipping noise, which allows us to explicitly control the ratio of noisy pairs in the dataset and quantify how different noise levels affect DPO optimization. As shown in Fig. 1, block (a) provides examples of noisy preference pairs sampled from the Pick-a-Pic v2 dataset (Kirstain

---

*Corresponding Author.

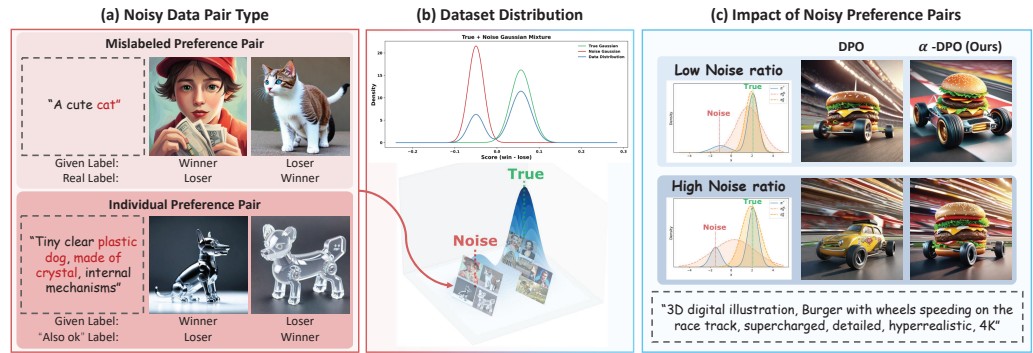

Figure 1: **Impact of noisy preference pairs on DPO.** Block (a) shows two types of noisy preference pairs from the Pick-a-Pic v2: mislabeled and individual pairs, which can be unified as label-flipping noise. Block (b) shows our estimated data distribution of the Pick-a-Pic V2 dataset. Block (c) illustrates that DPO is highly sensitive to such label-flipping noise, leading to degraded performance.

et al., 2023), and block (c) compares optimization under low and high noise ratios, illustrating how a larger proportion of noisy pairs substantially degrades the training efficacy of DPO.

To address this critical vulnerability of DPO to noisy data, previous research has focused on developing noise-robust DPO variants. They (Chowdhury et al., 2024; Fujisawa et al., 2025; Liang et al., 2024) typically rely on modifying loss functions or applying heuristics under simplified noise assumptions (*e.g.*, I.I.D. flip assumptions, known noise rates). These approaches often lack the fundamental ability to distinguish the true human preferences from spurious modes introduced by noisy data in real-world preference datasets. As shown in Fig. 1 block (b), the noisy data used in preference alignment shows a different distribution from the clean data. Moreover, these methods are primarily developed for Large Language Models (LLMs).

To better understand this limitation, we revisit DPO from a distributional perspective. In essence, DPO optimizes the model to align with the distribution of preferred (winner) samples while diverging from that of dispreferred (loser) ones. The presence of mislabeled preference pairs and individual preference pairs significantly compromises this process. Our theoretical analysis reveals that DPO's optimization objective is equivalent to minimizing the Forward Kullback–Leibler (FKL) divergence between the model distribution and the target preference distribution. However, FKL is inherently noise-sensitive due to its mass-covering behavior: it heavily penalizes underestimation even in regions where the target distribution assigns negligible probability. This property amplifies the influence of noisy or mislabeled samples, degrading alignment.

To achieve robust preference alignment in the presence of noise, two key criteria must be satisfied: (i) Mode-Seeking: Optimization should actively favor high-density regions over full support coverage, prioritizing the learning of salient preferences; (ii) Bounded Influence of Outliers: The loss function should be resilient to individual erroneous or outliers in the preference dataset. Motivated by these criteria, we introduce $\alpha$-DPO, which replaces the FKL with the more general $\alpha$-divergence, naturally promoting mode-seeking behavior and enhancing tolerance to noisy preferences. Crucially, since the optimal $\alpha$ depends on the underlying structure of these noisy data, we introduce a dynamic $\alpha$ scheduling mechanism that adaptively adjusts $\alpha$ based on the observed preference probability of each sample. This makes our approach inherently adaptive to preference deviations, leading to more reliable and robust alignment in practice. Extensive experiments showing that $\alpha$-DPO consistently surpasses existing noise-robust baselines across diverse benchmarks and noise regimes, affirming its effectiveness and generality. Comprehensive ablation studies further confirm the individual impact of each component in our framework.

Our contributions are summarized as follows: (i) We are the first to explore a noise-robust DPO method for image generation and theoretically prove Diffusion-DPO's inherent vulnerability to noisy preferences by demonstrating its optimization objective's equivalence to FKL divergence. (ii) We propose $\alpha$-DPO, which applies $\alpha$-Divergence to DPO to improve noise tolerance, and design a dynamic noise assessment method that adaptively adjusts its noise robustness during training for enhanced preference alignment. (iii) Extensive evaluations across synthetic and real-world datasets reveal that our $\alpha$-DPO exhibits robust noise tolerance, achieves superior preference alignment, and

consistently surpasses existing noise robust preference learning baselines, all without incurring additional computational cost.

## 2 RELATED WORK

**Preference Alignment of Diffusion Model.** Preference learning for human preference alignment has recently achieved remarkable advancement in diffusion image generation (Fan et al., 2023; Black et al., 2023; Yang et al., 2024; Zhu et al., 2025; Lu et al., 2025). Early explorations, such as DDPO (Black et al., 2023) and DPOK (Fan et al., 2023), utilize Reinforcement Learning (RL) algorithms to steer backbone models towards alignment with high-reward images (Xu et al., 2023; Kirstain et al., 2023; Wu et al., 2023). However, numerous studies indicate that these RL-based methods often suffer from reward hacking (Jena et al., 2025) and tend to yield suboptimal results. Another line of work leverages the DDIM sampling process and trains a step-wise reward model (Liang et al., 2025) for better preference alignment. However, these methods explicitly require reward model training. Alternatively, D3PO (Yang et al., 2024) directly uses binary human feedback to fine-tune the diffusion model. Similarly, Diffusion-DPO (Wallace et al., 2024) adopts the DPO algorithm (Rafailov et al., 2023) from Large Language Models (LLMs) (Han et al., 2024; Go et al., 2023), implicitly reparameterizing the reward model via the pre-trained model and the optimized model. Another large range of methods (Lu et al., 2025; Wang et al., 2025; Li et al., 2024; Hong et al., 2024; Shen et al., 2025; Yuan et al., 2024) builds on DPO to further improve the performance of preference alignment in Text-to-Image generation. However, these direct fine-tuning methods remain vulnerable to data noise, which can lead to a significant performance degradation.

**Divergence-Based Preference Optimization.** Several recent works explore divergence-based preference optimization (Lee et al., 2025; Gupta et al., 2025; Wu et al., 2024b; Santurkar et al., 2023; Song et al., 2023). AlphaPO (Gupta et al., 2025) leverages an $\alpha$-transform to reshape the reward model to stabilize preference optimization, while preserving the original KL-based DPO objective structure. FKPD (Shan et al., 2025) introduces forward-KL regularization into Diffusion-DPO to improve mode coverage and mitigate out-of-distribution generations in diffusion models. $\alpha$-DPO (Wu et al., 2024a) proposes a dynamic margin control version of IPO (Azar et al., 2024), aiming at enhancing the preference alignment when the reference model is not well-calibrated. In addition, $\alpha$-divergence has been applied as a policy regularizer in RLHF-style objectives, such as AlphaPO (Gupta et al., 2025) and $f$-DPO (Wang et al., 2023), by replacing the standard KL regularization term between the learned policy and the reference policy. In contrast to these approaches, our method directly reformulates DPO as a divergence-minimization objective between the learned preference distribution and the target preference distribution. Furthermore, a mode-seeking $\alpha$-divergence is adopted to improve robustness against noisy preference labels.

**Robust Direct Preference Optimization.** Following advances in learning with noisy labels in image classification (Song et al., 2022), A broad range of DPO variants have been developed to improve preference alignment under noisy feedback. These approaches can be broadly grouped into three categories: (i) Sample Selection methods (Kong et al., 2024; Deng et al., 2025; Cheng et al., 2024; Singh et al., 2024) filter out noisy samples by leveraging models trained on clean datasets. However, they are often limited by the requirement of clean data and the need to train multiple noise-validation models. (ii) Regularization-based methods, such as ROPO (Liang et al., 2024), assign regularized gradient weights according to label uncertainty to mitigate noise, while Conservative DPO (Mitchell, 2023) applies label smoothing. (iii) Loss modification methods improve robustness by optimizing loss. Robust-DPO (Chowdhury et al., 2024) introduces a theoretically grounded noise-robust loss design, but its impracticality demands prior knowledge of the noise ratio $\varepsilon$ is unfeasible in real-world scenarios. NAPO (Zhang et al., 2025) combines the noise robust Mean Absolute Error (MAE) (Ghosh et al., 2015) with Binary Cross-Entropy (BCE) to alleviate modality bias in MLLMs. More recently, Hölder-DPO (Fujisawa et al., 2025) evaluates the robustness using the statistics concept of redescending properties, revealing that no existing DPO variant satisfies this criterion. Its robustness, however, is derived under an I.I.D. flip assumption, which limits predictive reliability under structured noise, such as annotator bias or divergent group preferences. Critically, all of these methods are designed for autoregressive language models, whereas our approach focuses on diffusion models, offering fundamentally distinct inspiration for aligning the Markov chains.

## 3 METHOD

Our goal is to achieve robust preference alignment for diffusion models under noisy human feedback. In Sec. 3.1, starting from Reinforcement Learning from Human Feedback (RLHF) and its simplified variant Direct Preference Optimization (DPO), we extend DPO to diffusion models and show that its objective reduces to minimizing Forward Kullback–Leibler (FKL) divergence, which is intrinsically noise-sensitive. In Sec. 3.2, to overcome this limitation, we introduce $\alpha$-DPO, a reformulation based on $\alpha$-divergence that enhances robustness by balancing mass-covering and mode-seeking. Furthermore, we design a dynamic $\alpha$ scheduling strategy that adapts to data quality, ensuring consistent alignment under varying noise levels.

### 3.1 REFORMULATING DIFFUSION PREFERENCE ALIGNMENT AS FKL DIVERGENCE

We trace the evolution for preference alignment, from RLHF to DPO and its extension to diffusion models (*i.e.*, Diffusion-DPO), and show that the resulting Diffusion-DPO objective can be reformulated as a FKL divergence minimization problem.

**Reinforcement Learning from Human Feedback.** RLHF has emerged as a powerful paradigm for aligning generative models with human preferences. At its core, the generative model's policy $p_\theta(\boldsymbol{x}_0|\boldsymbol{c})$ is updated using reinforcement learning (e.g., PPO (Schulman et al., 2017)) to maximize the reward predicted by a reward function $r_\phi(\boldsymbol{c}, \boldsymbol{x}_0)$, with the optimization objective of:

$$\max_{p_\theta} \mathbb{E}_{\boldsymbol{c}\sim\mathcal{D}, \boldsymbol{x}_0\sim p_\theta(\boldsymbol{x}|\boldsymbol{c})} \left[r_\phi(\boldsymbol{c}, \boldsymbol{x})\right] - \beta \mathbb{D}_{KL}\left(p_\theta(\boldsymbol{x}_0|\boldsymbol{c})||p_{\text{ref}}(\boldsymbol{x}_0|\boldsymbol{c})\right). \tag{1}$$

Here, $p_{\text{ref}}(\boldsymbol{x}_0|\boldsymbol{c})$ is a fixed reference policy and $\beta$ is a hyperparameter controlling the FKL.

**Direct Preference Optimization.** The solution to the above objective Eq. 1 can be written as:

$$p^* = \frac{1}{Z(c)} p_{\text{ref}} \exp(\beta^{-1} r_\phi(\boldsymbol{c}, \boldsymbol{x}_0)), \tag{2}$$

where $Z(c) = \sum_{\boldsymbol{x}} p_{\text{ref}}(\boldsymbol{x}_0|\boldsymbol{c}) \exp\left(\beta^{-1} r_\phi(\boldsymbol{c}, \boldsymbol{x}_0)\right)$ is the partition function. However, directly optimizing this objective is challenging, as it necessitates a separate reward model $r_\phi(\boldsymbol{c}, \boldsymbol{x}_0)$ and relies on computationally expensive, often unstable, algorithms for policy updates. To circumvent these complexities, DPO reparameterizes the reward function directly in terms of the policy, allowing for end-to-end optimization without an explicit reward model:

$$r(\boldsymbol{c}, \boldsymbol{x}_0) = \beta \log \frac{p_\theta(\boldsymbol{x}_0|\boldsymbol{c})}{p_{\text{ref}}(\boldsymbol{x}_0|\boldsymbol{c})} + \beta \log Z(\boldsymbol{c}), \tag{3}$$

DPO grounds its formulation in the Bradley-Terry (BT) model. Given a preference pair $(\boldsymbol{x}_0^w, \boldsymbol{x}_0^l, \boldsymbol{c})$, where $\boldsymbol{x}_0^w$ and $\boldsymbol{x}_0^l$ is the "winner" and "loser" samples, the probability of $\boldsymbol{x}_0^w \succ \boldsymbol{x}_0^l|\boldsymbol{c}$ is:

$$p_{\text{BT}}(\boldsymbol{x}_0^w \succ \boldsymbol{x}_0^l|\boldsymbol{c}) = \sigma\left(r(\boldsymbol{c}, \boldsymbol{x}_0^w) - r(\boldsymbol{c}, \boldsymbol{x}_0^l)\right), \tag{4}$$

where $\sigma$ is the sigmoid function. Substituting Eq. 3 into Eq. 4, DPO derives a direct loss function that trains the policy $p_\theta$ to satisfy these preferences:

$$\mathcal{L}_{\text{DPO}}(\theta) = -\mathbb{E}_{(\boldsymbol{x}_0^w, \boldsymbol{x}_0^l, \boldsymbol{c})\sim\mathcal{D}} \left[\log \sigma \left(\beta \log \frac{p_\theta(\boldsymbol{x}_0^w|\boldsymbol{c})}{p_{\text{ref}}(\boldsymbol{x}_0^w|\boldsymbol{c})} - \beta \log \frac{p_\theta(\boldsymbol{x}_0^l|\boldsymbol{c})}{p_{\text{ref}}(\boldsymbol{x}_0^l|\boldsymbol{c})}\right)\right]. \tag{5}$$

**Diffusion-DPO.** To apply DPO to diffusion models, the parameterized distribution $p_\theta(\boldsymbol{x}_0|\boldsymbol{c})$ is not tractable, therefore, Diffusion-DPO (Wallace et al., 2024) utilize the evidence lower bound (ELBO) and define $R(\boldsymbol{c}, \boldsymbol{x}_{0:T})$ as the reward on the whole diffusion trajectory $(\boldsymbol{x}_1, \ldots, \boldsymbol{x}_T)$:

$$r(c, \boldsymbol{x}_0) = \mathbb{E}_{p_\theta(\boldsymbol{x}_{1:T}|\boldsymbol{x}_0, \boldsymbol{c})}\left[R(\boldsymbol{c}, \boldsymbol{x}_{0:T})\right], \tag{6}$$

The FKL regularization term in Eq. 1 is replaced with its upper bound $\mathbb{E}_{\text{KL}}\left[p_\theta(\boldsymbol{x}_{0:T}|\boldsymbol{c})p_{\text{ref}}(\boldsymbol{x}_{0:T}|\boldsymbol{c})\right]$. Through algebraic manipulation, the objective of Diffusion-DPO is formulated as:

$$\mathcal{L}_{\text{DPO-Diffusion}}(\theta) = -\mathbb{E}_{(\boldsymbol{x}^w, \boldsymbol{x}^l)\sim\mathcal{D}} \log \sigma \left(\beta \mathbb{E}_{\substack{\boldsymbol{x}_{1:T}^w\sim p_\theta(\boldsymbol{x}_{1:T}^w|\boldsymbol{x}_0^w) \\ \boldsymbol{x}_{1:T}^l\sim p_\theta(\boldsymbol{x}_{1:T}^l|\boldsymbol{x}_0^l)}} \left[\log \frac{p_\theta(\boldsymbol{x}_{0:T}^w)}{p_{\text{ref}}(\boldsymbol{x}_{0:T}^w)} - \log \frac{p_\theta(\boldsymbol{x}_{0:T}^l)}{p_{\text{ref}}(\boldsymbol{x}_{0:T}^l)}\right]\right). \tag{7}$$



Figure 2: Qualitative DPO results on SDXL with different label flipping.

| Backbone | Flip rate | CLIP↑ | HPSv2↑ | PS↑ | IR↑ | Aes↑ |
|---|---|---|---|---|---|---|
| SD1.5 | Pretrained | 0.3142 | 25.06 | 20.66 | 0.09535 | 5.450 |
| | 10% | 0.3152 | 25.92 | 20.92 | 0.3031 | 5.586 |
| | 20% | 0.3146 | 25.86 | 20.89 | 0.3015 | 5.606 |
| | 30% | 0.3102 | 25.11 | 20.75 | 0.2205 | 5.468 |
| | 40% | 0.3143 | 24.92 | 20.62 | 0.1289 | 5.421 |
| SDXL | Pretrained | 0.3240 | 28.20 | 21.99 | 0.7234 | 5.932 |
| | 10% | 0.3312 | 29.43 | 22.38 | 0.9424 | 5.960 |
| | 20% | 0.3310 | 29.12 | 22.27 | 0.9102 | 5.940 |
| | 30% | 0.3260 | 28.50 | 22.05 | 0.8649 | 5.926 |
| | 40% | 0.3256 | 28.77 | 22.00 | 0.7308 | 5.921 |

Table 1: DPO results with different label flipping.

by approximating the reverse process with the forward process, this loss can finally be simplified to per-step alignment loss across the diffusion trajectory. This enables the diffusion model to be optimized directly on pairwise preference data.

**DPO Loss as FKL Divergence.** As shown before, the objective of preference optimization is to minimize the distances between $p_\theta$ and $p^*$. Therefore, we can explicitly define it as a distribution matching problem: Let $\bar{p}_\theta(\boldsymbol{x}_{0:T}|\boldsymbol{c}) \propto p_\theta(\boldsymbol{x}_{0:T}|\boldsymbol{c})^\beta \cdot p_{\text{ref}}(\boldsymbol{x}_{0:T}|\boldsymbol{c})^{1-\beta}$ and

$\bar{p}^*(\boldsymbol{x}_{0:T}|\boldsymbol{c}) \propto p_{\text{ref}} \exp(r(\boldsymbol{x}_{0:T}, \boldsymbol{c}))$. We have the Diffusion-DPO objective Eq. 7 as:

$$\mathcal{L}_{\text{DPO-Diffusion}} = \mathbb{E}_{\boldsymbol{x} \sim \mathcal{D}} \left[ \mathbb{D}_{\text{KL}} \left[ \bar{p}^*(\boldsymbol{x}_{0:T}|\boldsymbol{c}))||\bar{p}_\theta(\boldsymbol{x}_{0:T}|\boldsymbol{c})] \right] \right]. \quad (8)$$

This objective aims to optimize $\bar{p}_\theta(\boldsymbol{x}_{0:T}|c)$ towards $\bar{p}^*(\boldsymbol{x}_{0:T}|c)$ which in terms is equivalent with optimizing $p_\theta$ towards $p^*$. The full proof can be found in the Appendix. A.2.

### 3.2 $\alpha$-DPO: ROBUST DIFFUSION PREFERENCE ALIGNMENT VIA $\alpha$ DIVERGENCE

We first analyze why the FKL divergence underlying Diffusion-DPO is intrinsically sensitive to noisy preferences. Motivated by this limitation, we then introduce the $\alpha$-divergence, which provides a principled trade-off between mass-covering and mode-seeking behaviors. Building on this foundation, we reformulate Diffusion-DPO under $\alpha$-divergence to obtain a more noise-robust objective, termed $\alpha$-DPO. Finally, we propose a dynamic $\alpha$ scheduling strategy that adaptively adjusts the divergence parameter during training, further enhancing robustness to varying noise levels.

**Noise Sensitivity of FKL Divergence.** As illustrated in Fig. 3, the FKL divergence's "mass-covering" property is susceptible to noise. This vulnerability stems from its severe penalty for instances where $\bar{p}^*(\boldsymbol{x}_{0:T}|\boldsymbol{c})$ assigns negligible probability to regions where $\bar{p}_\theta(\boldsymbol{x}_{0:T}|\boldsymbol{c})$ has even a small mass. Experiments results can be found in Fig. 2 and Tab. 1. To address this, we introduce the $\alpha$-divergence (Eq. 9), which balances mass-covering and mode-seeking via $\alpha$, improving robustness to noisy data and enabling more targeted alignment (Fig. 3).

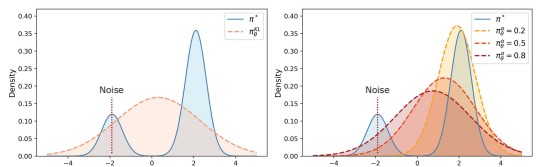

Figure 3: Distributions learned from noisy data (blue) under FKL divergence (left) and $\alpha$-divergence (right). The FKL tends to fit the noisy distribution closely, while $\alpha$-divergence yields a distribution closer to the noise-free case.

**$\alpha$-Divergence.** The $\alpha$-divergence offers a continuous spectrum of divergences that measures the difference between two distributions, parameterized by $\alpha \notin 0, 1$. For two probability distributions $\boldsymbol{P}$ and $\boldsymbol{Q}$, the $\alpha$-divergence is defined as:

$$\mathcal{D}_\alpha(\boldsymbol{P}||\boldsymbol{Q}) = \frac{1}{\alpha(\alpha-1)} \mathbb{E}_{x \sim \boldsymbol{Q}} \left[ \left( \frac{\boldsymbol{P}(x)}{\boldsymbol{Q}(x)} \right)^{1-\alpha} - (1-\alpha)\frac{\boldsymbol{P}(x)}{\boldsymbol{Q}(x)} - \alpha \right]. \quad (9)$$

as $\alpha \to 1$, $\alpha$-divergence simplifies to $\mathcal{D}_\alpha \to \mathcal{D}_{\text{KL}}(\boldsymbol{P}||\boldsymbol{Q})$, while $\alpha \to 0$, $\mathcal{D}_\alpha \to \mathcal{D}_{\text{KL}}(\boldsymbol{Q}||\boldsymbol{P})$. The hyperparameter $\alpha$ fundamentally controls the relative weighting of errors when $\boldsymbol{Q}(x)$ is small versus when $\boldsymbol{P}(x)$ is small, thereby influencing the divergence's mode-seeking or mass-covering behavior, which is crucial for tailoring divergence behavior in the presence of noise. Please see Fig. 3.

**$\alpha$-DPO: Preference Alignment via $\alpha$-Divergence.** Given these observations, we propose our noise-aware learning objective, $\alpha$-DPO. This optimization objective replaces the FKL with a more

noise-robust $\alpha$-divergence to align with general human preferences. According to Eq. 2, we can rewrite the reward model as $R(\boldsymbol{c}, \boldsymbol{x}_{0:T}) = \log \bar{p}_\theta^*(\boldsymbol{x}_{0:T}|\boldsymbol{c}) - \log p_{\text{ref}}(\boldsymbol{x}_{0:T}|\boldsymbol{c})$. By defining the relative log ratio of $\log \bar{p}_\theta(\boldsymbol{x}_{0:T}|\boldsymbol{c}) - \log p_{\text{ref}}(\boldsymbol{x}_{0:T}|c)$ as $g_\theta(\boldsymbol{c}, \boldsymbol{x}_{0:T})$, the target can be simplified as:

$$\mathbb{D}_\alpha\left[\bar{p}^* \parallel \bar{p}_\theta\right] = \mathbb{E}_{p_{\text{ref}}}\left[\frac{1}{\alpha(\alpha-1)}p^{g_\theta}\left((\frac{p^r}{p^{g_\theta}})^{1-\alpha} - (1-\alpha)\frac{p^r}{p^{g_\theta}} - \alpha\right)\right], \quad (10)$$

where $p^r = \frac{1}{Z_r(\boldsymbol{x}_{0:T})}\exp r(\boldsymbol{c}, \boldsymbol{x}_{0:T})$ and $p^{g_\theta(\boldsymbol{c}, \boldsymbol{x}_{0:T})} = \frac{1}{Z_{g_\theta}(\boldsymbol{c}, \boldsymbol{x}_{0:T})}\exp g_\theta(\boldsymbol{c}, \boldsymbol{x}_{0:T})$. However, the partition functions are intractable. By using the Monte Carlo estimation, we assume there are $K$ images for every prompt $\boldsymbol{c}$, and we can approximate the distribution of $p^r$ and $p^{g_\theta}$ as:

$$p^r(i|(\{\boldsymbol{x}_{0:T}\}_{1:K}, \boldsymbol{c})) = \frac{\exp\left(r(\boldsymbol{c}, \{\boldsymbol{x}_{0:T}\}_i)\right)}{\sum_{j=1}^K \exp\left(r(\boldsymbol{c}, \{\boldsymbol{x}_{0:T}\}_j)\right)}, \quad (11)$$

$$p^{g_\theta}(i|(\{\boldsymbol{x}_{0:T}\}_{1:K}, \boldsymbol{c})) = \frac{\exp\left(g_\theta(\boldsymbol{c}, \{\boldsymbol{x}_{0:T}\}_i)\right)}{\sum_{j=1}^K \exp\left(g_\theta(\boldsymbol{c}, \{\boldsymbol{x}_{0:T}\}_j)\right)}. \quad (12)$$

In our setting, the preference dataset is composed of pair-wise data $(\boldsymbol{x}_0^w, \boldsymbol{x}_0^l, \boldsymbol{c})$, i.e., $K = 2$. Thus, the final objective of $\mathcal{L}_{\alpha-\text{DPO}}$ can be written as:

$$\mathcal{L}_{\alpha\text{-DPO}} = \mathbb{E}_{\boldsymbol{c}\sim\mathcal{D}}\mathbb{E}_{\{\boldsymbol{x}_0^w, \boldsymbol{x}_0^l\}\sim p_{\text{ref}}}\left[\frac{1}{\alpha(\alpha-1)}u\cdot\left(u^{\alpha-1} - (1-\alpha)u^{-1} - \alpha\right)\right], \quad (13)$$

where $u = \sigma(g_\theta(\boldsymbol{c}, \boldsymbol{x}_{0:T}^w) - g_\theta(\boldsymbol{c}, \boldsymbol{x}_{0:T}^l))$. Following Diffusion-DPO (Wallace et al., 2024), recall that the $p_\theta(\boldsymbol{x}_{0:T}, \boldsymbol{c})$ is evaluated over the full diffusion trajectory $\boldsymbol{x}_{0:T}$, we apply Jensen's inequality to evaluate the upper bound of $p_\theta(\boldsymbol{x}_{0:T}, \boldsymbol{c})$ and approximating the reverse process $p_\theta(\boldsymbol{x}_{t-1}, \boldsymbol{x}_t|\boldsymbol{x}_0)$ using the forward distribution $q(\boldsymbol{x}_{1:T}|\boldsymbol{x}_0)$. This yields:

$$\mathcal{L}_{\alpha\text{-DPO}} = \mathbb{E}_{\substack{(\boldsymbol{x}_0^w, \boldsymbol{x}_0^l)\sim\mathcal{D}, t\sim\mathcal{U}(0,T), \\ \boldsymbol{x}_t^w\sim q(\boldsymbol{x}_t^w|\boldsymbol{x}_0^w), \\ \boldsymbol{x}_t^l\sim q(\boldsymbol{x}_t^l|\boldsymbol{x}_0^l)}}\left[\frac{1}{\alpha(\alpha-1)}u_t(\theta)\cdot\left(u_t^{\alpha-1}(\theta) - (1-\alpha)u_t^{-1}(\theta) - \alpha\right)\right], \quad (14)$$

the $u_t(\theta)$ is defined as:

$$u_t(\theta) = \sigma\Bigg(-\beta T\omega(\lambda_t)\bigg[\|\epsilon^w - \epsilon_\theta(\boldsymbol{x}_t^w, t)\|_2^2 - \|\epsilon^w - \epsilon_{\text{ref}}(\boldsymbol{x}_t^w, t)\|_2^2$$
$$- \left(\|\epsilon^l - \epsilon_\theta(\boldsymbol{x}_t^l, t)\|_2^2 - \|\epsilon^l - \epsilon_{\text{ref}}(\boldsymbol{x}_t^l, t)\|_2^2\right)\bigg]\Bigg). \quad (15)$$

with $\epsilon \sim \mathcal{N}(0, \mathbf{I})$, $t \sim \mathcal{U}(0, T)$, $\boldsymbol{x}_t^w \sim q(\boldsymbol{x}_t^w|\boldsymbol{x}_0^w)$, $\boldsymbol{x}_t^l \sim q(\boldsymbol{x}_t^l|\boldsymbol{x}_0^l)$, and $\lambda_t = \frac{\gamma_t^2}{\sigma_t^2}$ is the signal to noise ratio. $\omega(\lambda_t)$ is the fixed weighting function. $T$ is the maximum training timesteps. The full derivation can be found in the Appendix A.2.

**Dynamic $\alpha$ Scheduling.** The $\alpha$-divergence, parameterized by $\alpha$, encompasses a broad spectrum of divergence measures. A crucial characteristic of this family is its heightened sensitivity to discrepancies in the distributional tails as the absolute value of $\alpha$ increases. This inherent property also implies that the divergence exhibits varying sensitivities to noise, depending on the chosen $\alpha$. Consequently, relying on a fixed $\alpha$ value may not consistently yield optimal performance, particularly when confronting diverse noise levels. To address this limitation, we propose a novel mechanism for dynamically adjusting the $\alpha$ value. Our approach integrates information about the current

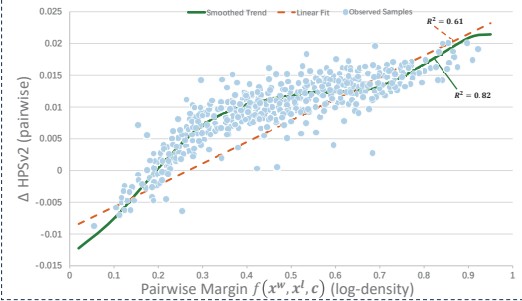

Figure 4: Comparison the value of $f(\boldsymbol{x}^w, \boldsymbol{x}^l, \boldsymbol{c})$ against the $\Delta$HPSv2 score. $f(\boldsymbol{x}^w, \boldsymbol{x}^l, \boldsymbol{c})$ exhibits a strong monotonic correlation with $\Delta$HPSv2, and could act as an effective internal confidence signal for guiding the dynamic adjustment of $\alpha$.

sample's potential noise level to guide this adjustment. To achieve this, we introduce the function $f(\boldsymbol{x}^w, \boldsymbol{x}^l, \boldsymbol{c}) = \text{Stop\_Grad}(u_t(\theta))$, which does not participate in the gradient backpropagation process, as an auxiliary indicator to quantify the noise level computed from each preference sample.

$f(\boldsymbol{x}^w, \boldsymbol{x}^l, \boldsymbol{c})$ *as an implicit preference classifier.* Given our target as minimizing the Eq. 14, we analyze the behaviour of the gradient of $\mathcal{L}_{\alpha\text{-DPO}}$ with respect to $u_t(\theta)$. The gradient is calculated as:

$$\nabla_{u_t}\mathcal{L}_{\alpha\text{-DPO}} = \frac{1}{(\alpha-1)}(u_t^{\alpha-1}-1) = \frac{1}{(1-\alpha)}(1-\frac{1}{u_t^{1-\alpha}}). \tag{16}$$

Given that the constrains are $0 < \alpha < 1$ and $0 < u_t < 1$, we have $\nabla_{u_t}\mathcal{L}_{\alpha\text{-DPO}} < 0$. This indicates that $\nabla_{u_t}\mathcal{L}_{\alpha\text{-DPO}}$ decreases monotonically. Consequently, to minimize the loss $\mathcal{L}_{\alpha\text{-DPO}}$, the model is guided to maximize the $u_t(\theta)$ ($f(\boldsymbol{x}^w, \boldsymbol{x}^l, \boldsymbol{c})$), thereby effectively increasing the likelihood of the preferred sample ($\boldsymbol{x}^w$) over the rejected sample ($\boldsymbol{x}^l$). This indicates that the auxiliary indicator $f(\boldsymbol{x}^w, \boldsymbol{x}^l, \boldsymbol{c})$ could act as an implicit preference classifier, where its value is analogous to a confidence score, with a higher score indicating a superior preference alignment. Furthermore, we validate this implicit classification role by comparing the value of $f(\boldsymbol{x}^w, \boldsymbol{x}^l, \boldsymbol{c})$ against the reference $\Delta$HPSv2 score in each sample pair (Singh et al., 2024). As shown in Fig. 4, $f(\boldsymbol{x}^w, \boldsymbol{x}^l, \boldsymbol{c})$ exhibits a strong monotonic correlation with $\Delta$HPSv2, confirming that the $f(\boldsymbol{x}^w, \boldsymbol{x}^l, \boldsymbol{c})$ is suitable for act as an effective internal confidence signal for guiding the dynamic adjustment of $\alpha$.

Therefore, we leverage the prediction of the fine-tuned model to simultaneously specify $\alpha$ as: $\alpha = \mu f(\boldsymbol{x}^w, \boldsymbol{x}^l, \boldsymbol{c})$ where $\mu$ is a hyper-parameter controlling the scale of $\alpha$. This process ensures that $\alpha$ is dynamically adapted: a higher $\alpha$ is assigned when samples achieve a larger confidence score, thereby indicating a less noisy sample. This adaptive strategy allows our model to maintain robust performance across varying data quality.

## 4 EXPERIMENTS

Table 2: Quantitative comparison with other backbones on Label Flipping dataset. The Flip rate is 20%. For more results, please refer to the Appendix. A.4.2. The best results are in highlighted **bold** and the second-best ones are underlined (same in the following tables).

| Dataset | Metric | SD1.5 | | | | | | SDXL | | | | | |
|---|---|---|---|---|---|---|---|---|---|---|---|---|---|
| | | Pretrained | DPO | cDPO | rDPO | H-DPO | Ours | Pretrained | DPO | cDPO | rDPO | H-DPO | Ours |
| Pick-a-Pic Test | CLIP↑ | 0.3142 | 0.3146 | 0.3139 | 0.3155 | 0.3112 | **0.3183** | 0.3240 | 0.3310 | 0.3247 | 0.3278 | 0.3304 | 0.3312 |
| | HPSv2↑ | 25.06 | 25.86 | 25.75 | 25.51 | 25.62 | **26.69** | 28.20 | 29.12 | 28,83 | 28.77 | 29.12 | **30.38** |
| | PS↑ | 20.66 | 20.89 | 20.89 | 20.90 | 20.51 | **21.00** | 21.99 | 22.27 | 22.14 | 22.17 | 22.29 | **22.50** |
| | IR↑ | 0.09535 | 0.3015 | 0.2393 | 0.1916 | 0.2629 | **0.4512** | 0.7234 | 0.9102 | 0.8568 | 0.8519 | 0.9211 | **1.001** |
| | Aes↑ | 5.450 | 5.606 | 5.618 | 5.589 | 5.604 | **5.623** | 5.932 | 5.940 | 5.936 | 5.925 | 5.937 | **5.961** |
| Pick-a-Pic Validation | CLIP↑ | 0.3086 | 0.3122 | 0.3100 | 0.3089 | 0.3073 | **0.3132** | 0.3197 | 0.3249 | 0.3228 | 0.3236 | 0.3251 | **0.3270** |
| | HPSv2↑ | 24.99 | 25.98 | 25.66 | 25.48 | 25.57 | **26.54** | 28.02 | 29.35 | 28.57 | 28.54 | 29.45 | **30.18** |
| | PS↑ | 20.60 | 20.94 | 20.80 | 20.81 | 20.84 | **20.98** | 21.89 | 22.24 | 22.08 | 22.09 | 22.28 | **22.39** |
| | IR↑ | 0.1035 | 0.2642 | 0.2127 | 0.1861 | 0.2617 | **0.3281** | 0.6770 | 0.9013 | 0.8095 | 0.8279 | 0.9157 | **0.9450** |
| | Aes↑ | 5.478 | 5.567 | 5.600 | 5.568 | 5.573 | **5.602** | 5.933 | 5.963 | 5.970 | 5.945 | 5.921 | **5.971** |
| HPSv2 | CLIP↑ | 0.3252 | 0.3214 | 0.3211 | 0.3202 | 0.3192 | **0.3256** | 0.3386 | 0.3436 | 0.3425 | 0.3419 | 0.3423 | **0.3438** |
| | HPSv2↑ | 24.68 | 25.49 | 25.31 | 25.56 | 25.21 | **26.88** | 28.26 | 29.75 | 29.01 | 29.01 | 29.85 | **30.64** |
| | PS↑ | 20.91 | 20.43 | 20.12 | 20.29 | 20.32 | **21.28** | 22.53 | 22.81 | 22.70 | 22.72 | 22.64 | **22.98** |
| | IR↑ | 0.1585 | 0.2913 | 0.2812 | 0.2720 | 0.3104 | **0.4492** | 0.8681 | 1.002 | 0.9754 | 0.9900 | 1.012 | **1.122** |
| | Aes↑ | 5.634 | 5.647 | 5.652 | 5.631 | 5.604 | **5.698** | 6.082 | 6.083 | 6.062 | 6.073 | 6.064 | **6.098** |

Table 3: Comparison with other baselines with models trained on the Pick-a-Pic V2 dataset.

| Dataset | Metric | SD1.5 | | | | | | SDXL | | | | | |
|---|---|---|---|---|---|---|---|---|---|---|---|---|---|
| | | Pretrained | DPO | cDPO | rDPO | H-DPO | Ours | Pretrained | DPO | cDPO | rDPO | H-DPO | Ours |
| Pick-a-Pic Test | CLIP↑ | 0.3142 | 0.3176 | 0.3158 | 0.3150 | 0.3133 | **0.3184** | 0.3240 | 0.3323 | 0.3321 | 0.3315 | 0.3322 | **0.3326** |
| | HPSv2↑ | 25.06 | 26.11 | 26.88 | 26.77 | 25.73 | **27.59** | 28.20 | 29.77 | 30.12 | 30.38 | 29.97 | **30.86** |
| | PS↑ | 20.66 | 21.02 | 21.19 | 21.18 | 20.73 | **21.27** | 21.99 | 22.43 | 22.41 | 22.46 | 22.25 | **22.51** |
| | IR↑ | 0.09535 | 0.3050 | 0.3806 | 0.3637 | 0.3369 | **0.5831** | 0.7234 | 0.9725 | 1.006 | 1.030 | 1.026 | **1.054** |
| | Aes↑ | 5.450 | 5.596 | 5.678 | 5.663 | 5.559 | **5.702** | 5.932 | 5.960 | 5.934 | 5.962 | 5.844 | **5.972** |
| Pick-a-Pic Validation | CLIP↑ | 0.3086 | 0.3131 | 0.3111 | 0.3105 | 0.3082 | **0.3152** | 0.3197 | 0.3286 | 0.3275 | 0.3271 | 0.3278 | **0.3289** |
| | HPSv2↑ | 24.99 | 26.24 | 26.70 | 26.61 | 25.44 | **27.29** | 28.02 | 29.73 | 30.02 | 30.10 | 29.49 | **30.64** |
| | PS↑ | 20.60 | 21.02 | 21.10 | 21.08 | 20.66 | **21.15** | 21.89 | 22.36 | 22.35 | 22.35 | 22.12 | **22.41** |
| | IR↑ | 0.1035 | 0.3191 | 0.3616 | 0.3506 | 0.3329 | **0.5337** | 0.6770 | 0.9235 | 0.9878 | 0.9837 | 0.9192 | **1.041** |
| | Aes↑ | 5.478 | 5.587 | 5.666 | 5.650 | 5.573 | **5.692** | 5.933 | 5.966 | 5.931 | 5.954 | 5.854 | **5.983** |
| HPSv2 | CLIP↑ | 0.3252 | 0.3292 | 0.3248 | 0.3246 | 0.3208 | **0.3302** | 0.3386 | 0.3442 | 0.3439 | 0.3436 | 0.3443 | **0.3451** |
| | HPSv2↑ | 24.68 | 25.89 | 26.65 | 26.55 | 25.75 | **27.68** | 28.26 | 30.05 | 30.53 | 30.68 | 30.22 | **31.42** |
| | PS↑ | 20.91 | 21.33 | 21.30 | 21.29 | 21.44 | **21.72** | 22.53 | 22.94 | 22.96 | 23.100 | 22.72 | **23.04** |
| | IR↑ | 0.1585 | 0.3628 | 0.4440 | 0.4299 | 0.4203 | **0.6391** | 0.8681 | 1.087 | 1.128 | 1.133 | 1.116 | **1.178** |
| | Aes↑ | 5.634 | 5.749 | 5.748 | 5.742 | 5.704 | **5.782** | 6.082 | 6.094 | 6.112 | 6.119 | 6.051 | **6.140** |

### 4.1 EXPERIMENT SETUP

**Dataset and Baselines.** Following Diffusion-DPO (Wallace et al., 2024), we select Stable Diffusion v1.5 (SD1.5) and Stable Diffusion XL Base 1.0 (SDXL) as the base models. We adopt Diffusion-DPO (Wallace et al., 2024) and robust alignment methods: conservative DPO (cDPO) (Mitchell,

2023), robust DPO (rDPO) (Chowdhury et al., 2024) and Hölder-DPO (H-DPO) (Fujisawa et al., 2025) for comparison. We train all the methods on the Pick-a-Pic v2 (Kirstain et al., 2023) Dataset. After excluding the $\sim 12\%$ of pairs with ties, the final training data is composed of 851293 pairs of preference data, with 58960 unique prompts.

**Implementation Details.** Training is distributed across 8 NVIDIA H100 80GB GPUs, with each GPU having a local batch size of 4, and we set the gradient accumulation over 64 iterations to achieve the desired global batch size 2048 (same as Diffusion-DPO). As models are trained with square resolutions (512 for SD1.5 and 1024 for SDXL) with $\mu$ set to 0.9999. A linear warmup of 200 steps is used for all methods. Finally, for $\alpha$-DPO trained on SD1.5, we set $\beta = 2000$ with a learning rate of $4.2 \cdot 10^{-8}$. For $\alpha$-DPO trained on SDXL, we set $\beta = 4000$ with a learning rate of $1.7 \cdot 10^{-8}$. We train all the baseline methods following their official guidelines. For more details of the baselines, please refer to the Appendix. A.3.

**Evaluation Details.** We evaluate the performance of our $\alpha$-DPO from two perspectives: (1) **Label Flipping**: We conduct experiments on a dataset where $\varepsilon$ of the labels are flipped, $\varepsilon$ is the flipping probability. These validate the effectiveness of our method compared with baselines in noise-awareness. (2) **Real World Benchmark**: Despite rigorous multi-round cleansing, current real-world datasets inevitably retain substantial amounts of noisy data. Therefore, we evaluate our proposed method and existing baselines on the real-world dataset, i.e., Pick-a-Pic V2, to demonstrate our approach's superior ability to effectively mitigate the influence of noise, highlighting its significant practical relevance. To ensure a fair comparison, we fix all the inference settings, where the guidance scale is 7.5 for SD1.5, 5 for SDXL, and the sampling step is 50. We include five automatic evaluation metrics to evaluate the performance of $\alpha$-DPO on text-image alignment, general preference, and aesthetic preference: (1) CLIP (Radford et al., 2021), (2) HPS V2 (Wu et al., 2023), (3) Pick Score (Kirstain et al., 2023), (4) ImageReward (Xu et al., 2023), (5) and Aesthetic Score (Schuhmann et al., 2022). For final testing, we generate images using captions from the HPSv2 (3200 captions) benchmark, PartiPrompt (Yu et al., 2022), along with 500 unique test captions and 500 unique validation captions sourced from the Pick-a-Pic V2 dataset.

## 4.2 EVALUATION ON SYNTHETIC DATASET

We started by testing our method's effectiveness on synthetic label-flipped samples. To do this, we randomly selected a specific proportion of training samples and inverted their preference labels. This synthetic dataset was then used to fine-tune SD1.5 and SDXL models with our proposed $\alpha$-DPO and other baseline methods. This preliminary experiment allows us to verify our method's robustness towards non-prevailing preferences. We rigorously evaluated our method and existing approaches across varying label flipping rates of 0.1, 0.2, 0.3, and 0.4. As shown in Tab. 2, we show quantitative results when setting the flipping rate to be 0.2 (More results please refer to the Appendix. A.4). The results demonstrated that our $\alpha$-DPO consistently yields significantly stronger performance compared to models fine-tuned with standard DPO at the same label-flipped level. Furthermore, our method outperforms cDPO and H-DPO across a variety of metrics. While rDPO achieves excellent performance in the noisy domain, our method surpasses it by a substantial margin. This suggests that the characteristics of non-prevailing preferences differ from the simplified noise models often assumed, highlighting the unique advantages of our approach.

**Winning Rate with different $\varepsilon$.** As shown in Tab. 4, our $\alpha$-DPO consistently outperforms all other methods across the CLIP, HPS V2, Pick Score, and ImageReward metrics. With a flipping rate of $\varepsilon = 0.2$, our method maintains a high winning rate of $82.6\%$ on HPS V2 and $76.6\%$ on Pick Score. In contrast, methods such as H-DPO and rDPO experience significant performance degradation in most test cases. Even at a higher flipping rate of 0.4, $\alpha$-DPO continues to outperform SDXL, achieving a $70.6\%$ winning rate on HPS V2 and $62.0\%$ on ImageReward. In comparison, most other methods fail to exceed $50\%$ winning rate against SDXL.

## 4.3 EVALUATION ON REAL-WORLD DATASET

While experiments with synthetic flipped labels offer initial insights, they do not fully capture the real-world value of our method due to fundamental differences between the distribution of synthetically generated noise and that of label flipping data arising from actual annotation processes. To address this, we directly fine-tune pretrained models on the Pick-a-Pic V2 dataset using our proposed

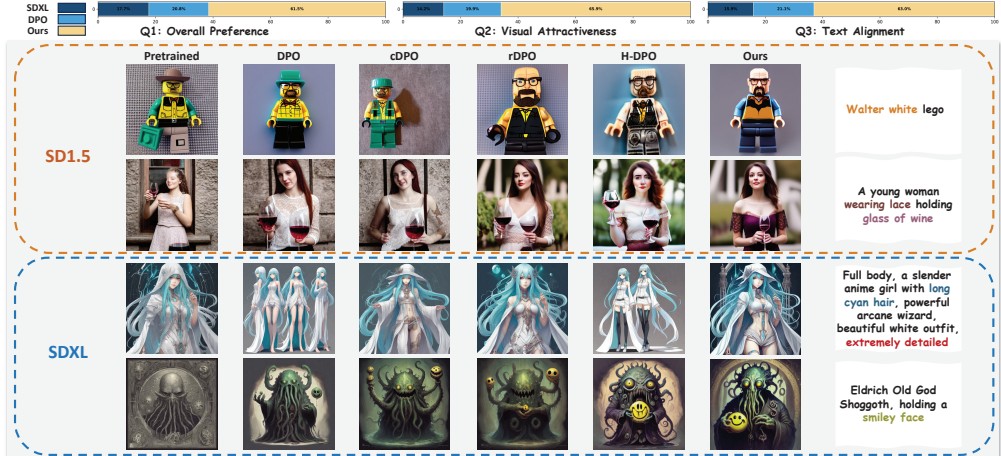

Figure 5: Top, human evaluations, our method shows superior performance over DPO on SDXL. Bottom, qualitative comparison with other baselines with models trained on Pick-a-Pic V2 Dataset.

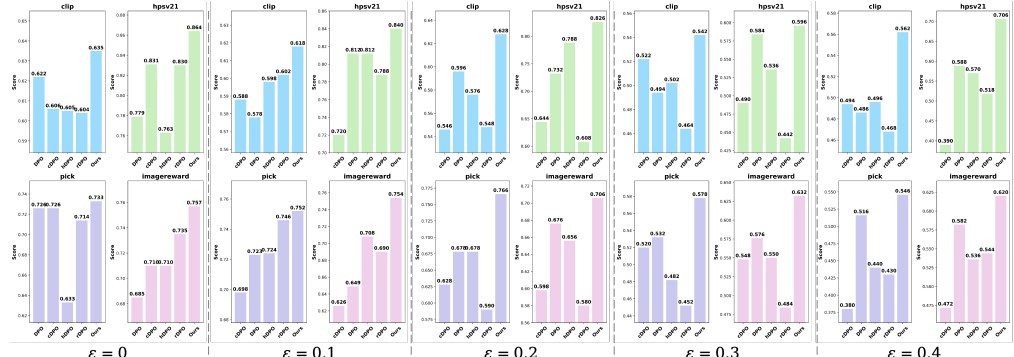

Figure 6: Winning rate (%) comparison for all methods versus SDXL on Pick-a-Pic V2 Test Dataset. From left to right, the winning rate of models trained on Pick-a-Pic V2 Dataset with label flipping rate $\varepsilon = 0, 0.1, 0.2, 0.3, 0.4$.

$\alpha$-DPO and various comparison methods. The results are presented in Tab. 3. Our method achieves better text-image alignment, general preference alignment, and aesthetic preference alignment.

For clearer comparison, in Fig. 5, we present a qualitative comparison of generated images from various methods and backbones. Our approach consistently yields superior image quality compared to Diffusion-DPO, a finding corroborated by our quantitative results. Specifically, when utilizing the SD1.5 backbone, our fine-tuned model generates correct semantics with significantly richer details in both human subjects and backgrounds. Furthermore, for the SDXL backbone, our method demonstrates enhanced performance over other techniques, showcasing improved text alignment alongside superior aesthetic qualities. This qualitative assessment unequivocally highlights the advanced capabilities and robustness of our proposed method.

### 4.4 ABLATION STUDY

We conduct ablation studies to evaluate the effect of the $\mu$, $\alpha$, and the proposed **Dynamic $\alpha$ Scheduling** on SDXL. Experimental results are shown in Tab. 4.

**Ablation on Parameter $\mu$.** The value of $\mu$ decides the robustness of our $\alpha$-DPO against noise. We choose $\mu \in 0.9999, 0.8, 0.5, 0.3, 0.1$ for the ablation study. We observe that as $\mu$ decreases, the alignment objective would overly emphasize the alignment of main modes, neglecting some other important information, which weakens the alignment precision.

**Ablation of the Dynamic $\alpha$ Scheduling.** Our **Dynamic $\alpha$ Scheduling** adaptively adjusts the noise-aware parameter $\alpha$. As evidenced in Tab. 4, in the 2nd row, disabling this strategy results in a substantial performance degradation.

Table 4: Ablation on hyperparameters $\mu$ and $\beta$ and the Dynamic $\alpha$ Strategy, evaluated on the Pick-a-Pic Test dataset with the fine-tuned SDXL model. (1) **Effect of** $\mu$: As $\mu$ grows, model performance first increases then decreases. (2) **Effect of the Dynamic** $\alpha$ **Strategy (Fixed-$\alpha$)**: Without dynamic allocation of the $\alpha$, the model performance degrades significantly. (3) **Effect of dynamic** $\alpha$ **start step**: the increasing of starting step slightly decrease the performance. (4) **Effect of** $\beta$: As $\beta$ increases, model performance first increases and then decreases.

| $\mu$ | 0.9999 | 0.8 | 0.5 | 0.3 | 0.1 | | 0.9999 | 0.8 | 0.5 | 0.3 | 0.1 |
|---|---|---|---|---|---|---|---|---|---|---|---|
| PS | 22.51 | 22.45 | 22.42 | 22.36 | 22.31 | IR | 1.054 | 1.032 | 1.025 | 1.021 | 1.013 |
| Fixed-$\alpha$ | 0.9999 | 0.8 | 0.5 | 0.3 | 0.1 | | 0.9999 | 0.8 | 0.5 | 0.3 | 0.1 |
| PS | 22.45 | 22.41 | 22.42 | 22.31 | 22.29 | IR | 1.019 | 1.018 | 1.014 | 1.002 | 0.9933 |
| dynamic $\alpha$ starts | 0 | 50 | 100 | 150 | 200 | | 0 | 50 | 100 | 150 | 200 |
| PS | 22.51 | 22.46 | 22.45 | 22.47 | 22.44 | IR | 1.054 | 1.047 | 1.048 | 1.041 | 1.031 |
| $\beta$ | 2000 | 3000 | 4000 | 5000 | 6000 | | 2000 | 3000 | 4000 | 5000 | 6000 |
| PS | 22.28 | 22.38 | 22.51 | 22.43 | 22.31 | IR | 1.022 | 1.031 | 1.054 | 1.042 | 1.002 |

**Ablation on the starting step of Dynamic $\alpha$ Scheduling.** To investigate the impact of when to activate the proposed Dynamic $\alpha$ Scheduling, we conduct an ablation study by varying the starting step at which the scheduling replaces the fixed $\alpha$. Specifically, we evaluate the model performance when Dynamic $\alpha$ Scheduling is introduced at the 0-th, 50-th, 100-th, 150-th, and 200-th training steps. This allows us to analyze how the timing of the transition from fixed to dynamic $\alpha$ influences the final results. The results are shown in Tab. 4. As the dynamic scheduling start step increases, the model performance exhibits a slight decline.

**Ablation on Parameter $\beta$.** As the quantitative results demonstrated in Tab. 4, starting from $\beta = 4000$, increasing $\beta$ makes the KL penalty become overly restrictive, limiting the model's optimization. On the other side, as $\beta$ decreases, the model learn to overly emphasize the reward, leading to a performance drop.

**DPO Variant Integration.** Fundamentally, our $\alpha$-DPO is formulated as a robust enhancement to the core DPO optimization mechanism, designed specifically to improve resilience against data noise. This architectural compatibility allows $\alpha$-DPO to serve as a "plug-and-play" module that

Table 5: Combining our $\alpha$-DPO with other DPO variants. SPO* denotes our reimplementation results. Our $\alpha$-DPO acts as a complementary component, which successfully integrates with SPO and significantly boosts its efficacy.

| Metrics | CLIP↑ | HPSv2↑ | PS↑ | IR↑ | Aes↑ |
|---|---|---|---|---|---|
| SPO | 0.3178 | 0.3227 | **22.89** | 1.107 | 6.366 |
| SPO* | 0.3119 | 0.3208 | 22.86 | 1.116 | 6.417 |
| SPO + Ours | **0.3201** | **0.3231** | **22.89** | **1.134** | **6.501** |

can be seamlessly integrated into other DPO-based frameworks. To demonstrate this versatility, we examine methods such as SPO (Liang et al., 2025), which enhance alignment by incorporating an auxiliary Reward Model (RM) prior into standard DPO training (conceptualized as RM + DPO). By substituting the conventional DPO component in these frameworks with our method, we construct an enhanced training paradigm: RM + $\alpha$-DPO. We present numerical comparisons on SDXL in Tab. 5, which confirm that this integration yields distinct performance gains over the original baselines. These results validate that $\alpha$-DPO is not only effective as a standalone method but also capable of generalizing to and significantly boosting the effectiveness of other DPO variants.

## 5 CONCLUSION

In this work, we investigated the vulnerability of existing DPO formulations to noisy human feedback, showing that their reliance on FKL divergence makes them intrinsically sensitive to noise. To address this challenge, we introduced $\alpha$-DPO, which reformulates preference alignment through $\alpha$-divergence and thereby achieves a principled balance between mass-covering and mode-seeking behaviors. Furthermore, we proposed a dynamic $\alpha$ scheduling mechanism that adapts to the estimated noise level of each sample, providing stronger robustness during training. $\alpha$-DPO achieves superior reliability in preference alignment for diffusion models across both synthetic and real-world datasets over previous baselines.

## 6 ETHICS STATEMENT

We acknowledge the ethical considerations associated with the development and application of text-to-image models, especially in the context of preference alignment. Our work prioritizes the alignment of these models to ensure that generated images reflect socially beneficial, fair, and inclusive preferences. We are mindful of the potential for bias in generative models and the harm that could arise from their misuse. To mitigate these risks, we have focused on approaches that promote positive, non-harmful applications of the technology.

## 7 REPRODUCIBILITY STATEMENT

We have made substantial efforts to ensure the reproducibility of our work. All novel algorithms and theoretical results are thoroughly explained in the main text, with detailed proofs provided in the Appendix. A.2. For experiments, we include comprehensive descriptions of the computational resources, parameter settings, and experimental protocols in both the main text and Appendix. A.3. Additionally, the source code and datasets used in this study, along with any relevant data processing steps, will be made available at Appendix. A.3 to support reproducibility. We encourage readers to refer to these resources to replicate our results.

## 8 ACKNOWLEDGMENTS

This work is supported by the National Key Research and Development Program of China under Grant No. 2024YFF0907202 and the National Natural Science Foundation of China (NSFC) under Grant 62372452, Jiangsu Provincial Science and Technology Major Project under Grant BG2024042, and the Joint Research Project on Integration of Culture and Science and Technology between Chinese Academy of Sciences and Hunan Province(#2024JK4003).

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

# A  APPENDIX

This supplementary material provides additional details as follows.

A.1. Use of Large Language Models.

A.2. Proofs and Derivations. We present a theoretical analysis of all the theorems and our method.

A.3. Experimental Details. We provide experimental details for the baselines.

A.4. More Analysis & Future Work. We provide more quantitative and qualitative results for $\alpha$-DPO fine-tuned on different base models and discuss the future work of $\alpha$-DPO.

### A.1 USE OF LARGE LANGUAGE MODELS

Large Language Models (LLMs) were used solely for minor grammar correction and stylistic polishing of the manuscript text. They were not involved in the design of the methodology, execution of experiments, analysis of results, or any other aspect of the scientific contribution.

### A.2 PROOFS AND DERIVATIONS

#### A.2.1 PROOFS OF EQ. 9

In this section, we provide a detailed proof of Eq. 9, showing that DPO's optimization objective is equivalent to minimizing the forward KL Divergence. Following $f$-PO Han et al. (2024), starting from the definition of the DPO loss and combining it with Eq. 3, we have:

$$
\begin{aligned}
\mathcal{L}_{\text{DPO}}(\theta) = \mathbb{E}_{\boldsymbol{c} \sim \mathcal{D}} \mathbb{E}_{p_{\text{ref}}(\{\boldsymbol{x}_{0:T}\}_{1:K})} & \left[ -\sum_{i=1}^{K} \frac{\exp\left(r(\boldsymbol{c}, \{\boldsymbol{x}_{0:T}\}_i)\right)}{\sum_{j=1}^{K} \exp\left(r(\boldsymbol{c}, \{\boldsymbol{x}_{0:T}\}_j)\right)} \right. \\
& \left. \cdot \log\left( \frac{\exp\left(\beta \log \frac{p_\theta(\{\boldsymbol{x}_{0:T}\}_i|\boldsymbol{c})}{p_{\text{ref}}(\{\boldsymbol{x}_{0:T}\}_i|\boldsymbol{c})}\right)}{\sum_{j=1}^{K} \exp\left(\beta \log \frac{p_\theta(\{\boldsymbol{x}_{0:T}\}_j|\boldsymbol{c})}{p_{\text{ref}}(\{\boldsymbol{x}_{0:T}\}_j|\boldsymbol{c})}\right)} \right) \right],
\end{aligned}
\tag{17}
$$

Here, using the Monte Carlo estimation of Eq. 11 and Eq. 12, the distribution of the target and optimized distribution are:

$$
p^r(i|\{\boldsymbol{x}_{0:T}\}_{1:K}|\boldsymbol{c}) = \sum_{i=1}^{K} \frac{\exp\left(r(\boldsymbol{c}, \{\boldsymbol{x}_{0:T}\}_i)\right)}{\sum_{j=1}^{K} \exp\left(r(\boldsymbol{c}, \{\boldsymbol{x}_{0:T}\}_j)\right)},
\tag{18}
$$

$$
p^{r(\theta)}(i|\{\boldsymbol{x}_{0:T}\}_{1:K}|\boldsymbol{c}) = \log\left( \frac{\exp\left(\beta \log \frac{p_\theta(\{\boldsymbol{x}_{0:T}\}_i|\boldsymbol{c})}{p_{\text{ref}}(\{\boldsymbol{x}_{0:T}\}_i|\boldsymbol{c})}\right)}{\sum_{j=1}^{K} \exp\left(\beta \log \frac{p_\theta(\{\boldsymbol{x}_{0:T}\}_j|\boldsymbol{c})}{p_{\text{ref}}(\{\boldsymbol{x}_{0:T}\}_j|\boldsymbol{c})}\right)} \right),
\tag{19}
$$

$r(\theta)$ is the optimized distribution parameterized by $\theta$. With the definition of $\bar{p}_\theta(\boldsymbol{x}_{0:T}|\boldsymbol{c}) \propto p_\theta(\boldsymbol{x}_{0:T}|\boldsymbol{c})^\beta \cdot p_{\text{ref}}(\boldsymbol{x}_{0:T}|\boldsymbol{c})^{1-\beta}$, we can extend the dpo loss to:

$$
\begin{aligned}
\mathcal{L}_{\text{DPO}}(\theta) = \mathbb{E}_{\boldsymbol{c} \sim \mathcal{D}} \mathbb{E}_{p_{\text{ref}}(\{\boldsymbol{x}_{0:T}\}_{1:K})} & \left[ \sum_{i=1}^{K} \frac{\exp\left(r(\boldsymbol{c}, \{\boldsymbol{x}_{0:T}\}_i)\right)}{\sum_{j=1}^{K} \exp\left(r(\boldsymbol{c}, \{\boldsymbol{x}_{0:T}\}_j)\right)} \right. \\
& \left. \cdot \log\left( \frac{\frac{\hat{p}_\theta(\{\boldsymbol{x}_{0:T}\}_i|\boldsymbol{c})}{p_{\text{ref}}(\{\boldsymbol{x}_{0:T}\}_i|\boldsymbol{c})}}{\sum_{j=1}^{K} \frac{\hat{p}_\theta(\{\boldsymbol{x}_{0:T}\}_j|\boldsymbol{c})}{p_{\text{ref}}(\{\boldsymbol{x}_{0:T}\}_j|\boldsymbol{c})}} \right) \right],
\end{aligned}
\tag{20}
$$

when $K \to \infty$, we have $\sum_{i=1}^{K} \exp g_\theta(\{\boldsymbol{x}_{0:T}\}_i|\boldsymbol{c})/K = \mathbb{E}_{\text{ref}(\boldsymbol{x}_{0:T}|\boldsymbol{c})} \exp g_\theta(\boldsymbol{x}_{0:T}|\boldsymbol{c})$. Note that $g_\theta(\boldsymbol{c}, \boldsymbol{x}_{0:T}) = \log \bar{p}_\theta(\boldsymbol{x}_{0:T}|\boldsymbol{c}) - \log p_{\text{ref}}(\boldsymbol{x}_{0:T}|c)$, then we have:

$$
\sum_{i=1}^{K} \frac{\bar{p}_\theta(\{\boldsymbol{x}_{0:T}\}_i \mid \boldsymbol{c})}{p_{\text{ref}}(\{\boldsymbol{x}_{0:T}\}_i \mid \boldsymbol{c})} = K,
\tag{21}
$$

$$
\begin{aligned}
\sum_{i=1}^{K} \exp\left(r(\{\boldsymbol{x}_{0:T}\}_i \mid \boldsymbol{c})\right) &= \mathbb{E}_{p_{\text{ref}}(\boldsymbol{x}_{0:T}|\boldsymbol{c})} \left[\exp\left(r(\boldsymbol{x}_{0:T} \mid \boldsymbol{c})\right)\right] \\
&= K Z(\boldsymbol{c}),
\end{aligned}
\tag{22}
$$

here, $Z(\boldsymbol{c})$ is the partition function. Putting the above to the Eq. 20, we obtain:

$$
\begin{aligned}
\mathcal{L}_{\mathrm{DPO}}(\theta) &= \mathbb{E}_{\boldsymbol{c} \sim \mathcal{D}}\left[ - \sum_{\boldsymbol{x}_{0:T} \in \mathcal{D}} p_{\mathrm{ref}}(\boldsymbol{x}_{0:T} \mid \boldsymbol{c}) \frac{\exp(r(\boldsymbol{x}_{0:T}, \boldsymbol{c}))}{Z(\boldsymbol{c})} \right. \\
&\qquad \left. \cdot \log \frac{\bar{p}_\theta(\boldsymbol{x}_{0:T} \mid \boldsymbol{c})}{\bar{p}^*(\boldsymbol{x}_{0:T} \mid \boldsymbol{c})} \right] \\
&= \mathbb{E}_{\boldsymbol{c} \sim \mathcal{D}}\left[ - \sum_{\boldsymbol{x}_{0:T} \in \mathcal{D}} p^*(\boldsymbol{x}_{0:T} \mid \boldsymbol{c}) \log \frac{\bar{p}_\theta(\boldsymbol{x}_{0:T} \mid \boldsymbol{c})}{\bar{p}^*(\boldsymbol{x}_{0:T} \mid \boldsymbol{c})} \right] \\
&= \mathbb{E}_{\boldsymbol{c} \sim \mathcal{D}}\left[ D_{\mathrm{KL}}\left( \bar{p}_\theta(\boldsymbol{x}_{0:T} \mid \boldsymbol{c}) \,\|\, \bar{p}^*(\boldsymbol{x}_{0:T} \mid \boldsymbol{c}) \right) \right].
\end{aligned}
\tag{23}
$$

which finishes the proof.

### A.2.2 $\alpha$ DIVERGENCE IS NOT EQUIVALENT TO A WEIGHTED FKL DIVERGENCE

We illustrate the inequality by comparing the generator function $f(u)$ of these two divergences. The generator function of weighted FKL divergence is $f_{FKL}(u) \propto -w \cdot \log u$ (w is the reweighting factor), while for $\alpha$ divergence, its generator function is $f_\alpha(u) \propto \frac{(u^{1-\alpha} - (1-\alpha)u - \alpha)}{(\alpha(\alpha-1))}$ (Wang et al., 2023). Unlike ad-hoc FKL reweighting, the $\alpha$ divergence structurally reshapes the nonlinearity of the underlying generator function, fundamentally altering how deviations from the target density are penalised. For example, for two samples $x_1$, $x_2$ with corresponding $u_2 = u_1^2$, here $u_1$ and $u_2$ are the density of $x_1$ and $x_2$ respectively. And we have:

$$
-w \cdot \log u_1 = \frac{(u_1^{1-\alpha} - (1-\alpha)u_1 - \alpha)}{(\alpha(\alpha-1))},
\tag{24}
$$

we can obtain

$$
-w \cdot \log u_2 = -2w \cdot \log u_1 = 2\frac{(u_1^{1-\alpha} - (1-\alpha)u_1 - \alpha)}{(\alpha(\alpha-1))},
\tag{25}
$$

On the other hand,

$$
\frac{(u_2^{1-\alpha} - (1-\alpha)u_2 - \alpha)}{(\alpha(\alpha-1))} = \frac{(u_1^{2-2\alpha} - (1-\alpha)u_1^2 - \alpha)}{(\alpha(\alpha-1))},
\tag{26}
$$

When $w$ and $\alpha$ are fixed, the equality:

$$
2\frac{(u_1^{1-\alpha} - (1-\alpha)u_1 - \alpha)}{(\alpha(\alpha-1))} = \frac{(u_1^{2-2\alpha} - (1-\alpha)u_1^2 - \alpha)}{(\alpha(\alpha-1))}.
\tag{27}
$$

does not hold for a wide range of $u_1$. This illustrates that $\alpha$-divergence introduces a qualitatively different regularisation.

Conceptually, conventional DPO remains oversensitive to label noise due to the inherently mass-covering behaviour of the forward KL divergence. Even when applying reweighted KL formulations, this tendency persists, leaving the model vulnerable to interference from noisy data. In contrast, the $\alpha$-divergence inherently promotes mode-seeking behaviour, effectively reducing the impact of noisy labels. This fundamental distinction motivates replacing the divergence formulation itself, rather than merely adjusting the weight of the forward KL term.

### A.3 EXPERIMENTAL DETAILS

### A.3.1 DETAILS OF BASELINES

1. **Diffusion-DPO (Wallace et al., 2024).** We use their released official checkpoint.

2. **Conservative DPO (Mitchell, 2023).** We train the cDPO with a learning rate of $\frac{2000}{\beta} 2.048 \cdot 10^{-8}$ with a linear warmup of 200 steps. We set $\beta = 2000$ and the contamination ratio as 0.1 for SD1.5. For SDXL, we set $\beta = 5000$ and the contamination ratio as 0.1.

3. **Robust DPO (Chowdhury et al., 2024).** We train the rDPO with a learning rate of $\frac{2000}{\beta} 2.048 \cdot 10^{-8}$ with a linear warmup of 200 steps. We set $\beta = 2000$ and the contamination ratio as $0.1$ for SD1.5. For SDXL, we set $\beta = 5000$ and the contamination ratio as $0.1$.

4. **Hölder-DPO (Fujisawa et al., 2025).** We train the H-DPO with a learning rate of $\frac{2000}{\beta} 2.048 \cdot 10^{-8}$ with a linear warmup of 200 steps. We set $\beta = 2000$ and the $\gamma = 1$ for SD1.5. For SDXL, we set $\beta = 5000$ and $\gamma = 1$.

5. **Diffusion-KTO (Li et al., 2024).** We use their released official checkpoint.

6. **Diffusion-SPIN (Yuan et al., 2024).** We use their released official checkpoint.

7. **SPO (Liang et al., 2025).** We use their released official checkpoint and the released official code to retrain the model. For training the enhanced SPO + Our model on SDXL, we adhered to the standard official hyperparameters provided by SPO, with two specific exceptions: $\beta = 3$ and our $\alpha$-DPO introduced hyperparameter $\mu = 0.9999$.

### A.3.2 DETAILS OF DATASETS

1. **Pick-a-Pic V2 (Kirstain et al., 2023) Dataset.** Pick-a-Pic is a comprehensive dataset of over 500,000 real user preferences, derived from more than 35,000 unique prompts submitted via a web platform. Each entry includes a text prompt, two AI-generated images, and user feedback on their preferred image or cases with no clear preference. The collection features outputs from various generative models, such as Stable Diffusion 2.1, Dreamlike Photoreal 2.05, and Stable Diffusion XL.

2. **HPSv2 (Wu et al., 2023) Dataset.** Human Preference Score v2 (HPS v2) is a large-scale and comprehensive benchmark for text-to-image models. The dataset comprises 798,090 human preference choices on 430,060 pairs of images. This makes it one of the largest datasets of its kind. Its test set is made up of 3200 unique prompts, which are used to generate a controlled set of images for evaluation. The prompts in the test set are categorized into four distinct styles to ensure a comprehensive evaluation of a model's generative abilities: Animation, Concept-art, Painting, and Photo.

3. **PartiPrompt (Yu et al., 2022) Dataset.** PartiPrompts is a rich set of over 1600 prompts that can be used to measure model capabilities across various categories and challenge aspects.

### A.3.3 DETAILS OF USER STUDY

For the human evaluation, we randomly sampled a set of prompts and used each method to generate 40 groups of images. We then recruited 40 participants, who were asked to evaluate the generated images along three dimensions: *(1) Which image is your overall preferred choice? (2) Which image is more visually attractive? (3) Which image better matches the text description?*. In the human evaluation, we utilize a diverse and comprehensive set of prompts. The complete listing of prompts used to generate the images is:

1. Anime art featuring Hatsune Miku with symmetrical shoulders.

2. A lifelike anime girl in steel plate armor takes a selfie in a castle courtyard.

3. A Rika Furude plush toy.

4. An anime girl, masterpiece, good line art, trending in pixiv, ,.

5. a dog riding bicycle.

6. two girls-sisters in dark heavy techno armor with red-shine eyes, long black hair, futuristic helmet on head, full body view, gorgeous face, Cute face, fantasy-style, highly detailed, VFX, 4k, ultrarealistic photo, realistic-style photo.

7. two horses running in a field in the foggy day time.

8. Eldrich Old God Shoggoth, holding a smiley face.

9. nice lake in austria mountains.

10. dark elf armor of world of warcraft, mistique background, ultra detail, full body shot, full body portrait, detailed, professional photo, Cinematic lighting, Cinematic, Color Grading.

11. floral textile A letter.

12. a diamond ring on a girls hand.

13. Two fake looking giraffes are on display at an exhibit.

14. Spongebob wearing Rick Owens clothing in an avant garde fashion look on r/Streetwear.

15. Man at park.

16. Photorealistic beautiful African girl age 25.

17. Koa is a muscular adventurer with a rugged tan complexion, piercing blue eyes, and a weathered face. He sports a full beard and a wide-brimmed hat. For his latest adventure, he rides atop the back of a magnificent flying turtle with a sleek shell and majestic wings. The turtle soars through a stunning sky filled with fluffy white clouds and vibrant colors. Koa grips tightly to the turtle's shell.

18. a car that is made of wood.

19. full body, a slender anime girl with long cyan hair, powerful arcane wizard, beautiful white outfit, extremely detailed, realistic shading.

20. Walter white lego.

21. Danny DeVito appears in Jojo's Bizarre Adventur.

22. A hyper-realistic 3D render of the Doge meme featuring a shiba inu, portrayed in cinematic lighting.

23. Harry Potter book cover with Van Gogh-inspired design and visible book title.

24. Anthropomorphic orange crocodi.

25. Two cats playing chess on a tree branch.

26. She has a heart-shaped face, large expressive eyes, and soft, lustrous hair. Her figure is slender and graceful, and she moves with a quiet confidence and inner strength that commands attention. But what truly sets her apart is her inner beauty, which radiates from within and lights up any room she enters.

27. a stop sign with a large tree behind it.

28. Batman wearing metal gear armor with a cinematic, dramatic background.

29. A sign that says "PICK A PIC".

30. 3D digital illustration, Burger with wheels speeding on the race track, supercharged, detailed, hyperrealistic, 4K.

31. Spinger Spaniel liver and white.

32. a building sized muscular bear.

33. dinossaur rpg character, planet with the volcanic landscape, concept art, smooth fog, high quality, symmetry in objects.

34. A cat jumping for a toy.

35. A Sign that says: Free Candy!

36. cyberpunk mutant cute blond girl.

37. flying crocodile.

38. beautiful portrait of a young woman made of glossy glass skin surrounded with glowing birds.

39. pikachu in a pinstripe suit.

40. logo of a blue elephant, flat modern vector icon.

In Fig. 7, we leverage human evaluation to compare the generated results of our method and baselines. The results depict that our method shows superior performance on "Text Alignment", "Visual Attractiveness", and "Overall Preference". To ensure that our human evaluation is statistically robust, we additionally calculate pairwise comparisons of the human evaluation results among SDXL, DPO, and $\alpha$-DPO, and compute the corresponding p-values. The results in Tab. 6 reveal that our $\alpha$-DPO significantly outperforms both DPO ($p < 0.001$) and SDXL ($p < 0.001$) in terms of human preference. The small $p$-value ($p < 0.001$) indicates that the observed preference difference has less than a $0.1\%$ probability of occurring by random chance. This outcome strongly demonstrates the statistical robustness of the proposed method in human evaluation.

Table 6: Human Evaluation Robustness Analysis. This result strongly demonstrates the statistical robustness of the proposed method in human evaluation.

|  | SDXL vs DPO | SDXL vs $\alpha$-DPO | DPO vs $\alpha$-DPO |
|---|---|---|---|
| p | 1.000e+00 | 5.002e-09 | 2.961e-9 |

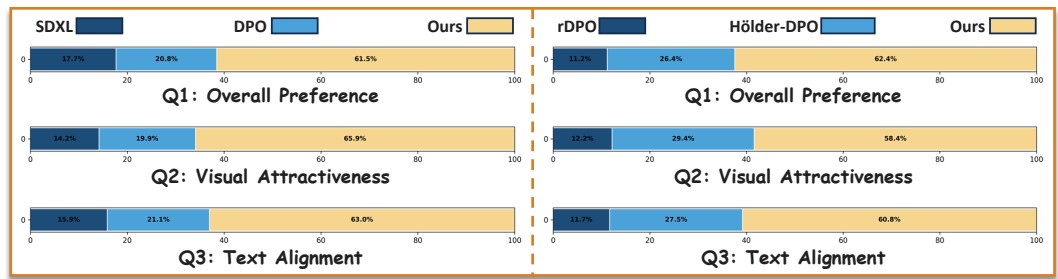

Figure 7: User Study. We conduct user studies to evaluate the performance of our method and the baselines. On the left block, we compare our $\alpha$-DPO with SDXL and DPO. On the right block, we compare our $\alpha$-DPO with rDPO and Hólder-DPO. Our method shows superior performance on "Text Alignment", "Visual Attractiveness", and "Overall Preference".

### A.3.4 COMPUTATION COST

We compare the computational cost of our method with the other baselines. Our method needs the same time and GPU consumption as the DPO and the other baselines. In Fig. 8, we show the training curve of them. Our method does not need extra training cost while showing better performance.

### A.3.5 EXPLAINATION OF PICKSCORE

As shown in Tab. 1, as the label-flipping rate increases from $10\%$ to $40\%$, PickScore exhibits a consistent downward trend. Although its absolute change appears small (*e.g.*, from 22.38 to 22.00), this is a property of the metric's scale rather than an indication of robustness. The qualitative degradation in the generated images, such as reduced text alignment and weaker visual appeal, is clearly visible in Fig. 2, which confirms that PickScore is correctly reflecting a decline in preference alignment, albeit within a narrow numeric range. To show that this limited PickScore change in amplitude does not signify robust performance, but rather an intrinsic failure of its scoring mechanism to be sensitive to underlying quality signals. Here, we leverage the Coefficient of Variation (CV) to analyze the property of PickScore. This analysis stems from an

Table 7: The Insufficient Amplitude Change in PickScore is a Symptom of its Low Intrinsic Sensitivity. The PickScore's low CV ($23.12\%$) demonstrate that the metric operates with a narrow scoring range. In contrast, ImageReward exhibits an extremely high CV ($188.24\%$), which indicates a sparse and highly selective scoring mechanism.

| Metrics | CV(%)-2000↑ | CV(%)-50000↑ |
|---|---|---|
| PickScore | 23.12 | 22.65 |
| ImageReward | 188.24 | 189.05 |

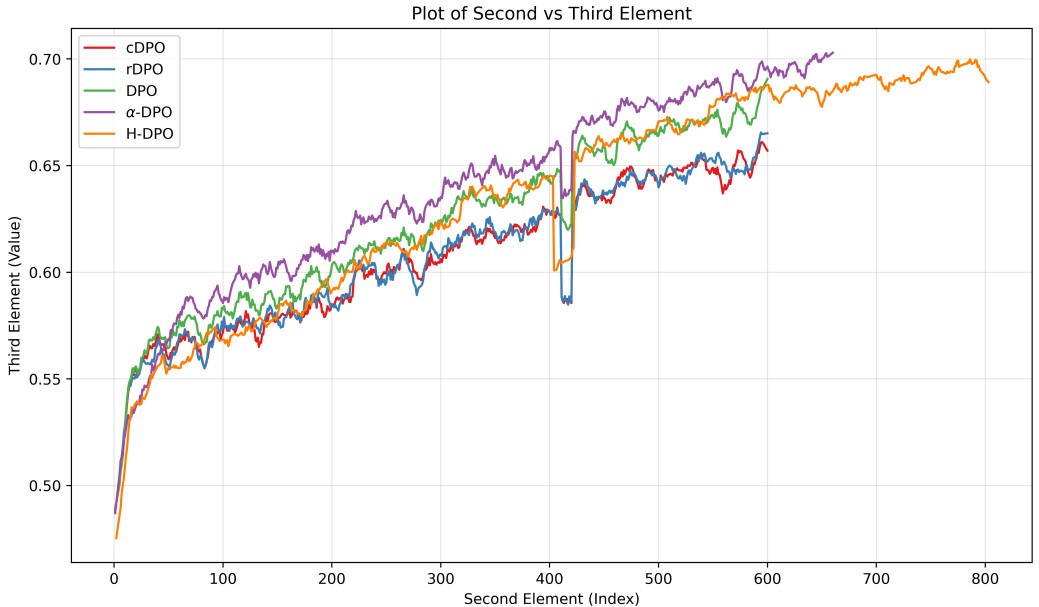

Figure 8: Training curve comparison of the used methods.

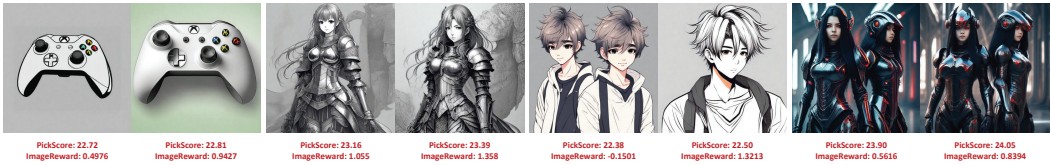

Figure 9: We show the PickScore and the ImageReward of 4 pairs of samples. Although PickScore exhibits smaller amplitude fluctuations, it follows the same trend as ImageReward.

experiment conducted on $2,000$ and $50000$ randomly selected image pairs from the Pick-a-Pic Test Dataset, where scores were normalized to compute the CV for comparative analysis. The results are depicted in Tab. 7. We have also provided visualizations of both PickScore and ImageReward outputs in Fig. 9, which offer immediate intuitive support for the following statistical observations: **(1) The PickScore's low CV ($23.12\%$) demonstrates that PickScore operates with a narrow scoring range relative to its mean. This results in an inherent insensitivity (low amplitude) to changes in sample quality (the observed small amplitude to the model's fundational scoring philosophy). (2) In contrast, ImageReward exhibits a significantly sharper score change under varied sample quality.**

## A.4 ADDITIONAL EXPERIMENTAL RESULTS

To further demonstrate the effectiveness of our method, we provide additional qualitative and quantitative evaluations with other methods.

### A.4.1 QUALITATIVE RESULTS

In Fig. 10, Fig. 11, Fig. 12, Fig. 13, Fig. 14 and Fig. 15, we provide more visualization results. Compared with other baselines, our fine-tuned model generates significantly richer details in both human subjects and backgrounds, showcasing improved text alignment alongside superior aesthetic qualities. This qualitative assessment unequivocally highlights the advanced capabilities and robustness of our proposed method.

### A.4.2 QUANTITATIVE RESULTS

**Other label flipping ratio.** We provide more experimental results of our method and the other baselines under label flipping rate $\varepsilon = 0.1$, $\varepsilon = 0.3$, and $\varepsilon = 0.4$ on SDXL. The results are demonstrated in Tab. 10, Tab. 11, and Tab. 12 respectively, our method consistently outperforms the other methods across a wide range of flipping rates. We also notice that, trained on dataset with $\varepsilon = 0.1$, our method even outperforms the Diffusion-DPO trained on $\varepsilon = 0$

**Comparison with other SOTA alignment methods.** To further validate the effectiveness of our method, we conducted a comprehensive comparison with additional state-of-the-art approaches for preference alignment on real-world datasets, as reported in Tab. 8 and Tab. 9. These baselines include Diffusion-KTO (Li et al., 2024), Diffusion-SPIN (Yuan et al., 2024), and MAPO (Hong et al., 2024). Our method consistently achieves superior preference alignment across metrics. In particular, on the Stable Diffusion 1.5 model, our approach significantly outperforms or matches all competing methods. This demonstrates that our method is not only highly robust to noise but also achieves superior alignment performance on real-world datasets compared to other dedicated alignment methods.

### A.4.3 FUTURE WORK & LIMITATIONS

(1) While $\alpha$-DPO demonstrates clear advantages in noise robustness and consistently surpasses existing baselines, it introduces an additional hyperparameter $\mu$ that controls the strength of dynamic $\alpha$-scheduling. Designing an automated or more adaptive mechanism for tuning $\mu$ represents a valuable direction for future work. (2) We have preliminarily explored combining $\alpha$-DPO with SPO, and we believe that developing more principled integration strategies in this direction offers substantial potential for future improvements.

Table 8: Quantitative comparison with other baselines with SD15 trained on the Pick-a-Pic V2 dataset.

| Dataset | Metric | SD1.5 | | | | |
| --- | --- | --- | --- | --- | --- | --- |
| | | Pretrained | DPO | KTO | SPIN | Ours |
| Pick-a-Pic Test | CLIP↑ | 0.3142 | 0.3176 | **0.3193** | 0.3095 | 0.3184 |
| | HPSv2↑ | 25.05 | 26.11 | 27.09 | **28.05** | 27.59 |
| | PS↑ | 20.66 | 21.02 | 21.13 | 21.18 | **21.27** |
| | IR↑ | 0.09535 | 0.3050 | **0.6123** | 0.4901 | 0.5831 |
| | Aes↑ | 5.596 | 5.678 | 5.700 | **5.937** | 5.702 |
| Pick-a-Pic Validation | CLIP↑ | 0.3086 | 0.3131 | **0.3160** | 0.3057 | 0.3152 |
| | HPSv2↑ | 24.99 | 26.24 | 27.02 | 27.33 | **27.29** |
| | PS↑ | 20.60 | 21.02 | 21.09 | **21.36** | 21.15 |
| | IR↑ | 0.1035 | 0.3191 | **0.6041** | 0.4632 | 0.5337 |
| | Aes↑ | 5.478 | 5.587 | 5.709 | **5.823** | 5.692 |
| HPSv2 | CLIP↑ | 0.3252 | 0.3292 | **0.3322** | 0.3205 | 0.3302 |
| | HPSv2↑ | 24.68 | 25.89 | **28.71** | 28.21 | 27.68 |
| | PS↑ | 20.91 | 21.33 | **21.44** | 21.32 | **21.44** |
| | IR↑ | 0.1585 | 0.3628 | 0.5321 | 0.5474 | **0.6391** |
| | Aes↑ | 5.634 | 5.749 | 5.712 | 5.702 | **5.782** |

Table 9: Quantitative comparison with other baselines with SDXL trained on the Pick-a-Pic V2 dataset.

| Dataset | Method | CLIP↑ | HPSv2↑ | PS↑ | IR↑ | Aes↑ |
|---------|--------|-------|--------|-----|-----|------|
| Pick-a-Pic Test | MaPO | 0.3251 | 29.18 | 22.07 | 0.853 | 5.932 |
| | Ours | **0.3326** | **30.86** | **22.51** | **1.054** | **5.972** |
| Pick-a-Pic Validation | MaPO | 0.3225 | 29.11 | 22.00 | 0.8596 | 5.874 |
| | Ours | **0.3289** | **30.64** | **22.41** | **1.041** | **5.983** |
| HPSv2 | MaPO | 0.3408 | 29.41 | 22.64 | 0.9844 | 6.072 |
| | Ours | **0.3451** | **31.42** | **23.04** | **1.178** | **6.140** |

Table 10: Quantitative comparison with other backbones on Label Flipping dataset with SDXL. The Flip rate is 10%. The best results are in highlighted **bold** and the second-best ones are underlined.

| Dataset | Metric | SDXL | | | | | |
|---------|--------|------------|------|------|------|-------|------|
| | | Pretrained | DPO | cDPO | rDPO | H-DPO | Ours |
| Pick-a-Pic Test | CLIP↑ | 0.3240 | 0.3312 | 0.3302 | 0.3307 | 0.3315 | **0.3320** |
| | HPSv2↑ | 28.20 | 29.43 | 29.51 | 29.85 | 29.69 | **30.57** |
| | PS↑ | 21.99 | 22.38 | 22.35 | 22.41 | 22.37 | **22.50** |
| | IR↑ | 0.7234 | 0.9424 | 0.9466 | 0.9641 | 0.9324 | **1.021** |
| | Aes↑ | 5.932 | 5.960 | 5.933 | 5.966 | 5.926 | **5.971** |
| Pick-a-Pic Validation | CLIP↑ | 0.3197 | 0.3272 | 0.3263 | 0.3269 | 0.3260 | **0.3273** |
| | HPSv2↑ | 28.02 | 29.53 | 29.41 | 29.67 | 29.64 | **30.33** |
| | PS↑ | 21.89 | 22.32 | 22.27 | 22.35 | 22.33 | **22.40** |
| | IR↑ | 0.6770 | 0.8973 | 0.8775 | 0.9137 | 0.9312 | **0.9710** |
| | Aes↑ | 5.933 | **5.963** | 5.924 | 5.961 | 5.915 | 5.962 |
| PartiPrompt | CLIP↑ | 0.3223 | 0.3280 | 0.3271 | 0.3277 | 0.3279 | **0.3281** |
| | HPSv2↑ | 27.24 | 28.69 | 28.73 | 28.93 | 29.12 | **29.87** |
| | PS↑ | 22.39 | 22.64 | 22.69 | 22.72 | 22.71 | **22.80** |
| | IR↑ | 0.7489 | 1.010 | 1.017 | 1.047 | 1.052 | **1.129** |
| | Aes↑ | 5.705 | 5.765 | 5.723 | 5.764 | 5.747 | **5.819** |
| HPSv2 | CLIP↑ | 0.3386 | 0.3437 | 0.3432 | 0.3440 | **0.3448** | 0.3444 |
| | HPSv2↑ | 28.26 | 29.95 | 29.82 | 30.07 | 30.02 | **30.99** |
| | PS↑ | 22.53 | 22.85 | 22.90 | 22.96 | 22.73 | **23.05** |
| | IR↑ | 0.8681 | 1.041 | 1.047 | 1.081 | 1.084 | **1.133** |
| | Aes↑ | 6.082 | 6.082 | 6.042 | 6.085 | 6.050 | **6.123** |

Table 11: Quantitative comparison with other backbones on Label Flipping dataset with SDXL. The Flip rate is 30%. The best results are in highlighted **bold** and the second-best ones are underlined.

| Dataset | Metric | SDXL | | | | | |
| --- | --- | --- | --- | --- | --- | --- | --- |
| | | Pretrained | DPO | cDPO | rDPO | H-DPO | Ours |
| Pick-a-Pic Test | CLIP↑ | 0.3240 | 0.3260 | 0.3253 | 0.3221 | 0.3259 | **0.3271** |
| | HPSv2↑ | 28.20 | 28.50 | 27.15 | 27.19 | 28.51 | **28.70** |
| | PS↑ | 21.99 | 22.05 | 21.97 | 21.90 | 21.99 | **22.12** |
| | IR↑ | 0.7234 | 0.8649 | 0.7862 | 0.7163 | 0.8871 | **0.9213** |
| | Aes↑ | 5.932 | 5.926 | 5.924 | 5.925 | 5.862 | **5.965** |
| Pick-a-Pic Validation | CLIP↑ | 0.3197 | 0.3222 | 0.3200 | 0.3192 | 0.3217 | **0.3247** |
| | HPSv2↑ | 28.02 | 28.35 | 27.91 | 27.83 | 28.38 | **28.62** |
| | PS↑ | 21.89 | 22.02 | 21.91 | 21.85 | 21.93 | **22.10** |
| | IR↑ | 0.6770 | 0.8037 | 0.7057 | 0.7006 | 0.8352 | **0.8762** |
| | Aes↑ | 5.933 | 5.921 | 5.932 | 5.957 | 5.894 | **5.973** |
| PartiPrompt | CLIP↑ | 0.3223 | 0.3245 | 0.3245 | 0.3238 | 0.3260 | **0.3272** |
| | HPSv2↑ | 27.24 | 27.64 | 27.24 | 27.15 | 27.54 | **27.83** |
| | PS↑ | 22.39 | 22.47 | 22.42 | 22.37 | 22.37 | **22.52** |
| | IR↑ | 0.7489 | 0.8459 | 0.7941 | 0.7642 | 0.8853 | **0.9046** |
| | Aes↑ | 5.705 | 5.716 | 5.681 | 5.681 | 5.662 | **5.789** |
| HPSv2 | CLIP↑ | 0.3386 | 0.3407 | 0.3392 | 0.3389 | 0.3422 | **0.3433** |
| | HPSv2↑ | 28.26 | 28.59 | 28.14 | 28.11 | 28.51 | **30.15** |
| | PS↑ | 22.53 | 22.62 | 22.52 | 22.48 | 22.54 | **22.71** |
| | IR↑ | 0.8681 | 0.9453 | 0.8911 | 0.8773 | 0.9723 | **1.005** |
| | Aes↑ | **6.082** | 6.024 | 6.038 | 6.040 | 6.002 | 6.078 |

Table 12: Quantitative comparison with other backbones on Label Flipping dataset with SDXL. The Flip rate is 40%. The best results are in highlighted **bold** and the second-best ones are underlined.

| Dataset | Metric | SDXL | | | | | |
| --- | --- | --- | --- | --- | --- | --- | --- |
| | | Pretrained | DPO | cDPO | rDPO | H-DPO | Ours |
| Pick-a-Pic Test | CLIP↑ | 0.3240 | **0.3256** | 0.3245 | 0.3247 | 0.3231 | 0.3254 |
| | HPSv2↑ | 28.20 | 28.77 | 27.75 | 28.28 | 28.73 | **29.55** |
| | PS↑ | 21.99 | 22.00 | 21.81 | 21.93 | 21.89 | **22.11** |
| | IR↑ | 0.7234 | 0.7308 | 0.7034 | 0.7592 | 0.7489 | **0.8530** |
| | Aes↑ | 5.932 | 5.921 | 5.941 | 5.930 | 5.941 | **5.963** |
| Pick-a-Pic Validation | CLIP↑ | 0.3197 | 0.3202 | 0.3185 | 0.3201 | 0.3207 | **0.3221** |
| | HPSv2↑ | 28.02 | 28.64 | 27.47 | 27.99 | 28.24 | **29.37** |
| | PS↑ | 21.89 | 21.93 | 21.78 | 21.83 | 21.87 | **22.04** |
| | IR↑ | 0.6770 | 0.7343 | 0.6462 | 0.6827 | 0.7240 | **0.8080** |
| | Aes↑ | 5.933 | 5.913 | 5.955 | **5.962** | 5.902 | **5.962** |
| PartiPrompt | CLIP↑ | 0.3223 | 0.3236 | 0.3244 | 0.3236 | 0.3234 | **0.3258** |
| | HPSv2↑ | 27.24 | 27.72 | 27.91 | 27.26 | 27.93 | **28.47** |
| | PS↑ | 22.39 | 22.43 | 22.41 | 22.36 | 22.38 | **22.45** |
| | IR↑ | 0.7489 | 0.8326 | 0.8482 | 0.7709 | 0.8007 | **0.8703** |
| | Aes↑ | 5.705 | 5.746 | **5.806** | 5.686 | 5.770 | 5.762 |
| HPSv2 | CLIP↑ | 0.3386 | 0.3383 | 0.3383 | 0.3384 | 0.3399 | **0.3405** |
| | HPSv2↑ | 28.26 | 28.32 | 27.95 | 28.46 | 28.43 | **29.89** |
| | PS↑ | 22.53 | 22.58 | 22.40 | 22.50 | 22.48 | **22.60** |
| | IR↑ | 0.8681 | 0.8912 | 0.8410 | 0.8785 | 0.9195 | **0.9627** |
| | Aes↑ | **6.082** | 6.046 | 6.056 | 6.067 | 6.042 | 6.072 |

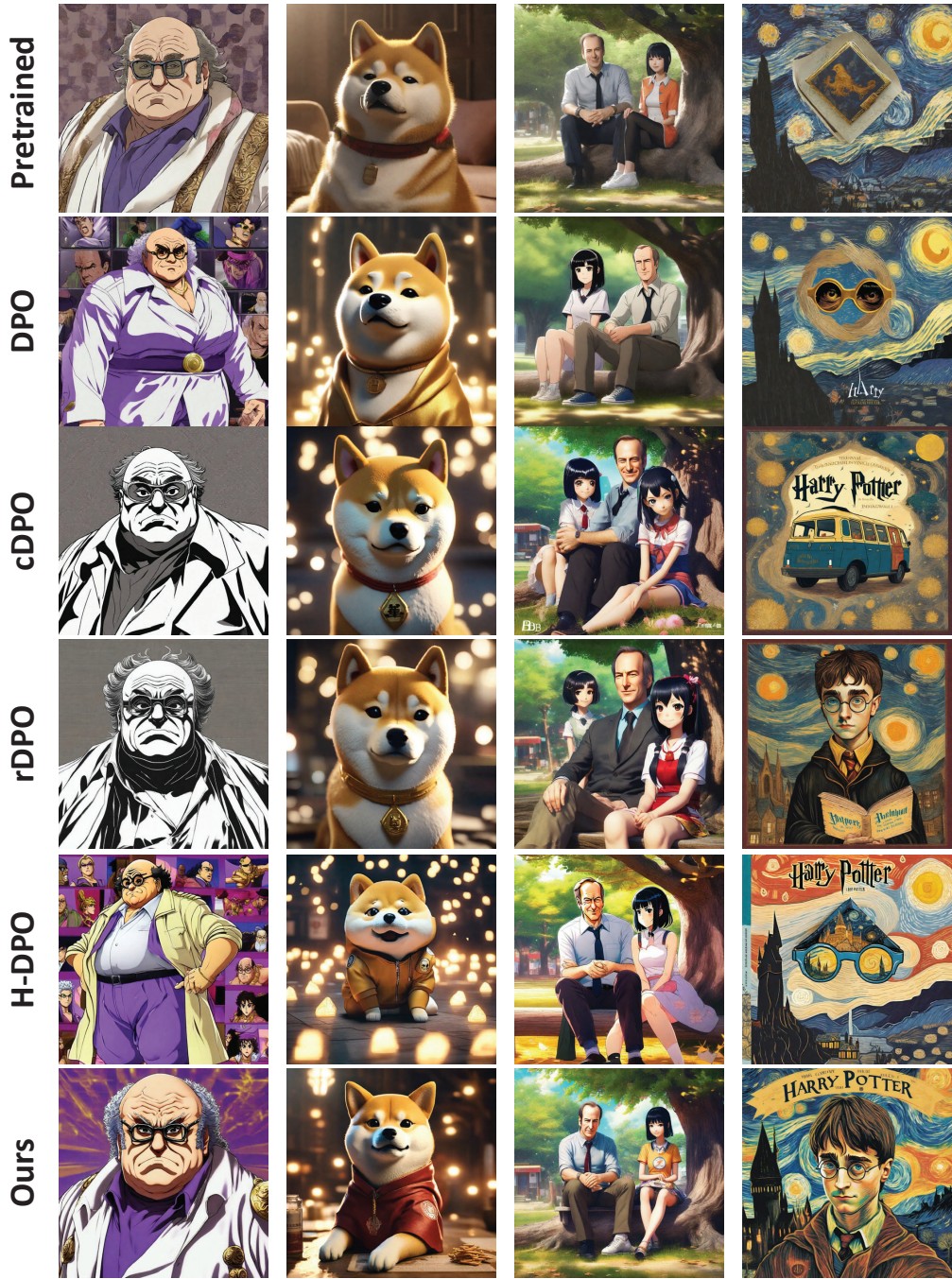

Figure 10: Images generated by different models (based on SDXL) for various prompts. From left to right, the prompts are as follows: 1. "Danny DeVito appears in Jojo's Bizarre Adventure.", 2. "A hyper-realistic 3D render of the Doge meme featuring a shiba inu, portrayed in cinematic lighting.", 3. "A photorealistic Bob Odenkirk is sitting under a tree with a smiling anime girl with black hair and hime cut in a digital art anime key visual.", 4. "Harry Potter book cover with Van Gogh-inspired design and visible book title.".

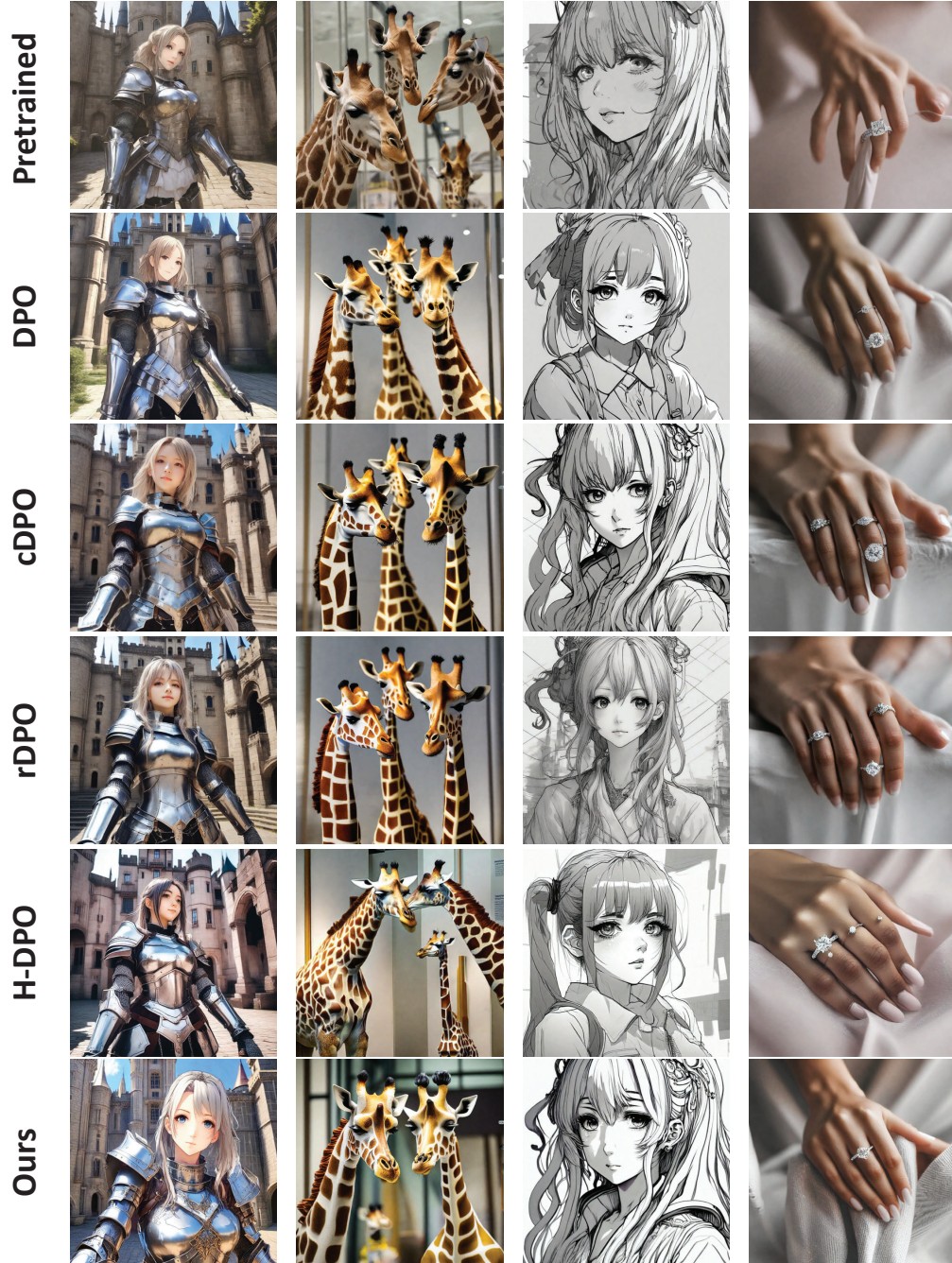

Figure 11: Images generated by different models (based on SDXL) for various prompts. From left to right, the prompts are as follows: 1. "A lifelike anime girl in steel plate armor takes a selfie in a castle courtyard", 2. "Two fake looking giraffes are on display at an exhibit", 3. "An anime girl, masterpiece, good line art, trending in pixiv,,", 4. "A diamond ring on a girls hand".

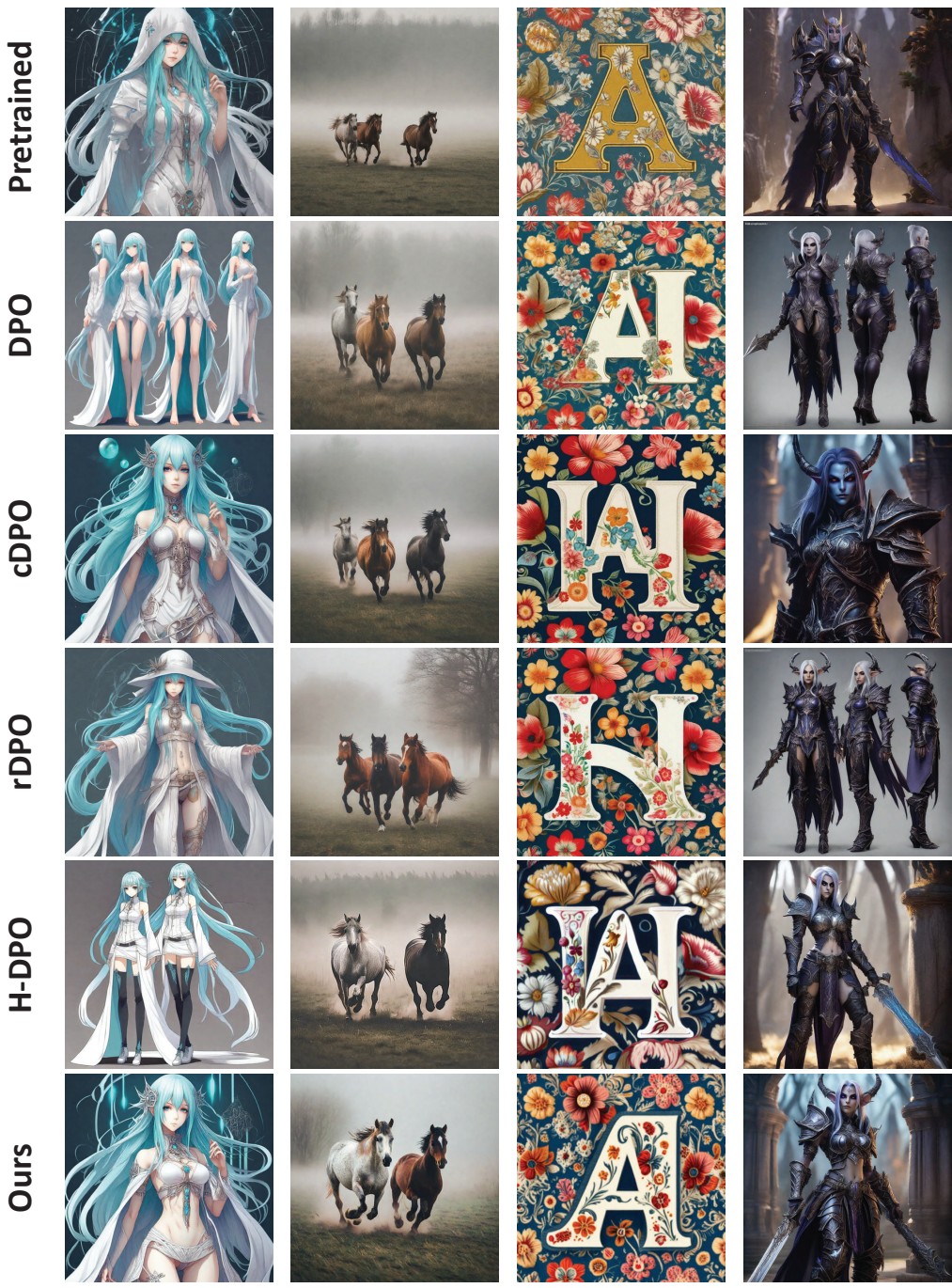

Figure 12: Images generated by different models (based on SDXL) for various prompts. From left to right, the prompts are as follows: 1. "full body, a slender anime girl with long cyan hair, powerful arcane wizard, beautiful white outfit, extremely detailed, realistic shading", 2. "two horses running in a field in the foggy day time", 3. "Floral textile A letter", 4. "dark elf armor of world of warcraft, mistique background, ultra detail, full body shot, full body portrait, detailed, professional photo, Cinematic lighting, Cinematic, Color Grading".

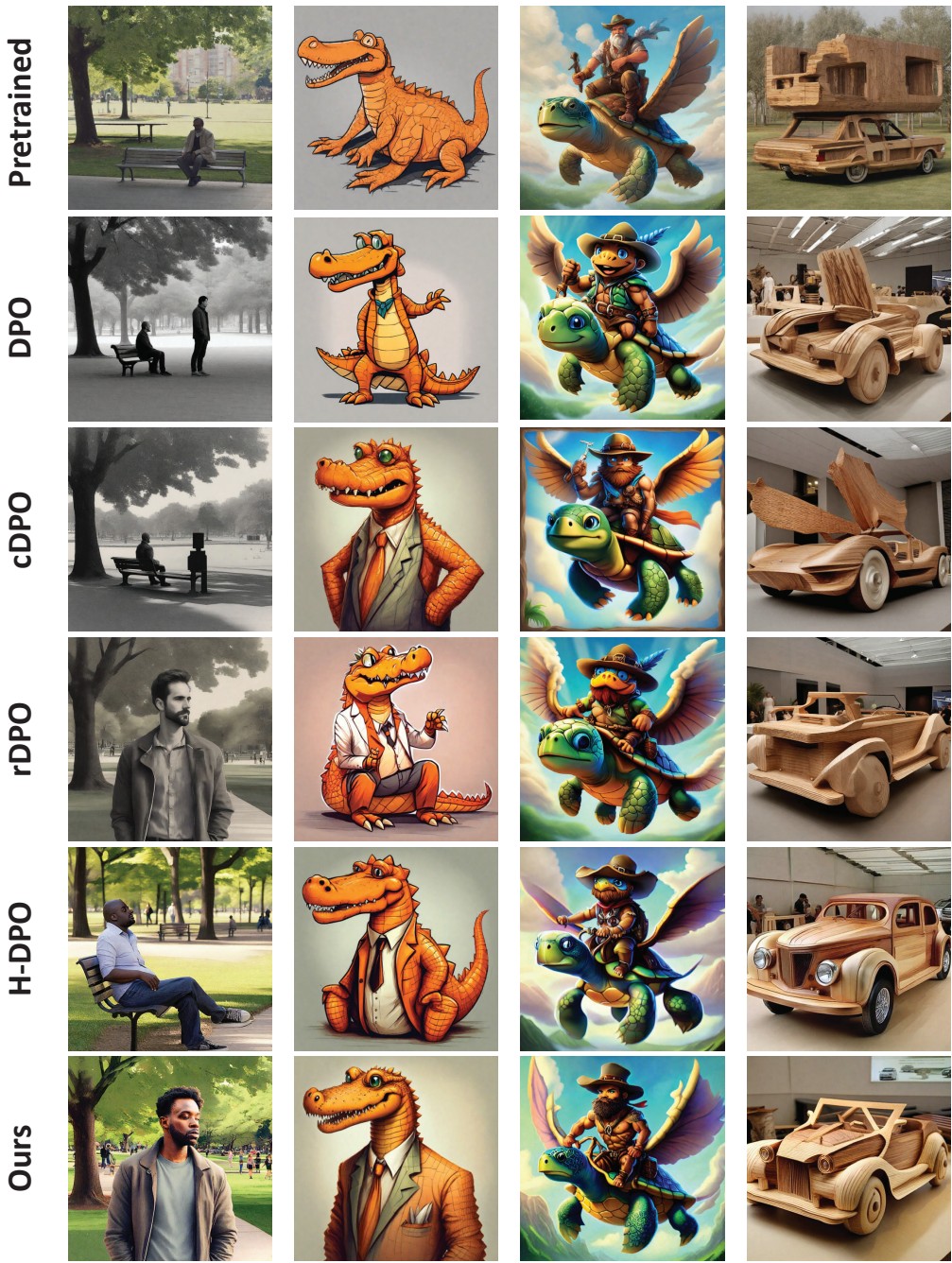

Figure 13: Images generated by different models (based on SDXL) for various prompts. From left to right, the prompts are as follows: 1. "Man at park", 2. "Anthropomorphic orange crocodile", 3. "Koa is a muscular adventurer with a rugged tan complexion, piercing blue eyes, and a weathered face. He sports a full beard and a wide-brimmed hat. For his latest adventure, he rides atop the back of a magnificent flying turtle with a sleek shell and majestic wings. The turtle soars through a stunning sky filled with fluffy white clouds and vibrant colors. Koa grips tightly to the turtle's shell.", 4. A car that is made of wood.".

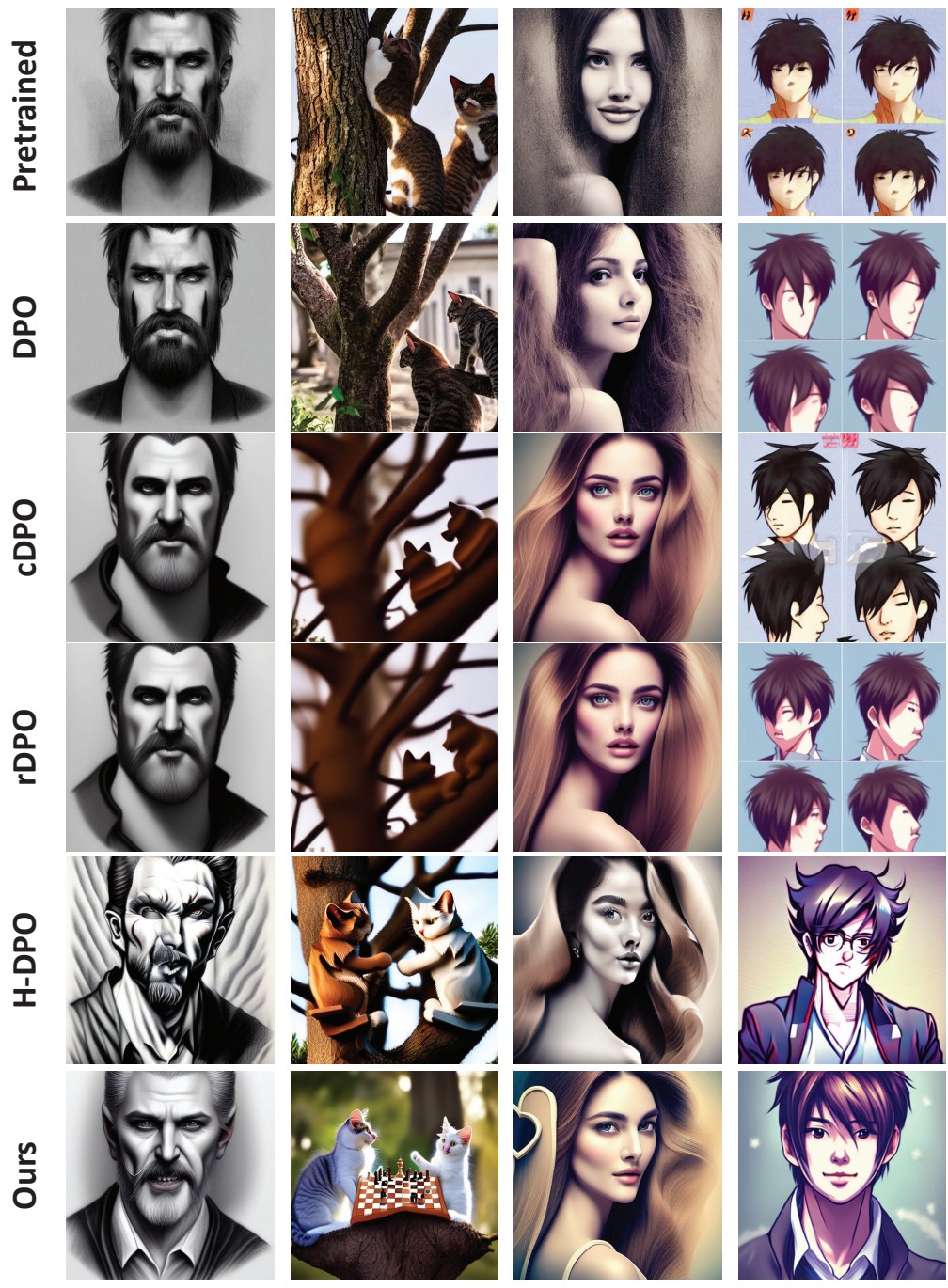

Figure 14: Images generated by different models (based on SD15) for various prompts. From left to right, the prompts are as follows: 1. "A pencil draw portrait of a grey haired 30 years old vampire with an extended goatee beard, trending on artstation.", 2. "Two cats playing chess on a tree branch.", 3. "She has a heart-shaped face, large expressive eyes, and soft, lustrous hair. Her figure is slender and graceful, and she moves with a quiet confidence and inner strength that commands attention. But what truly sets her apart is her inner beauty, which radiates from within and lights up any room she enters.", 4. "Cute guy, Asian popular pop hair style, Anime character.".

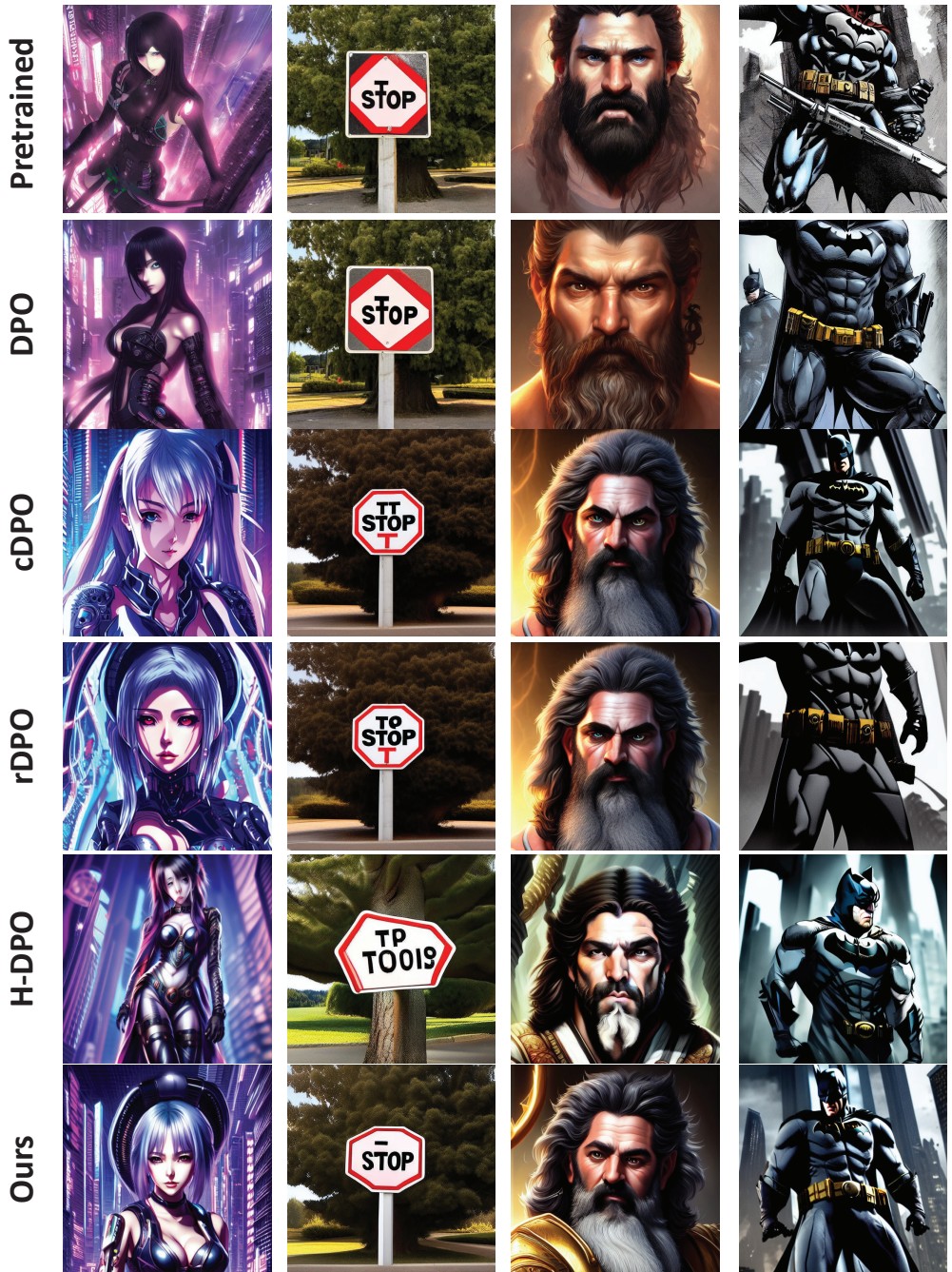

Figure 15: Images generated by different models (based on SD15) for various prompts. From left to right, the prompts are as follows: 1. "The image depicts an anime woman in a futuristic cyberpunk setting, with intricate and highly detailed elements crafted in digital painting.", 2. "a stop sign with a large tree behind it.", 3. "A highly detailed digital portrait of the mythological figure Hephaestus, featuring a fantasy interpretation inspired by D&D and well-known French rugby player Sebastien Chabal, as illustrated by artists Artgerm, Greg Rutkowski, and Magali Villeneuve and gaining popularity on ArtStation.", 4. "Batman wearing metal gear armor with a cinematic, dramatic background."

