# OpenReview forum: "$\alpha$-DPO: Robust Preference Alignment for Diffusion Models via $\alpha$ Divergence"
_ICLR.cc/2026/Conference — ICLR 2026 Poster_

### Official Review · Reviewer_kScQ · 2025-10-19

**Soundness:** 2
**Presentation:** 3
**Contribution:** 3
**Rating:** 6
**Confidence:** 4

**Summary:**

This paper proposed the modified DPO algorithm with $\alpha$-divergence for diffusion model alignment based on the learning dynamics analysis on the plain DPO's implicit reward. With simple modification, the authors test their method, $\alpha$-DPO, on varying axes of datasets, models, and metrics, consistently outperforming previous methods.

**Strengths:**

1. The empirical scope of the paper is extensive, with different models (SD 1.5 and SDXL), a lot of baselines (Diffusion-DPO, SPIN, KTO, …), and metrics/datasets.
2. The evaluation results demonstrate that $\alpha$-divergence is superior in nearly every case mentioned in the paper, shown through the direct win rate and average evaluation scores.
3. Although it is not a novel approach, he motivation for using $\alpha$-divergence is straightforward (see weakness for the details).

**Weaknesses:**

The main weakness of the paper lies in the **novelty and completeness of the divergence analysis on DPO and $\alpha$-divergence as the solution, which serves as the conceptual foundation of the work**. The paper’s discussion of DPO through the lens of divergence minimization and the proposed adoption of $\alpha$-divergence as a solution overlooks several important prior works that have already analyzed these aspects in detail:

1. Theoretical analyses of DPO as divergence minimization (Section 3.1) [2, 4].

2. Adoption of $\alpha$-divergence in DPO-style formulations (Section 3.2) [1–3].

A few previous studies, highly relevant to this paper’s core idea, have already investigated how different divergences articulate the implicit reward under the DPO framework, including the explicit use of $\alpha$-divergence [1–3]. While [1] and [2] directly address $\alpha$-divergence and its policy-based variants, [3] generalizes DPO to a broader family of $f$-divergences that naturally subsume $\alpha$-divergence. None of these works is cited or discussed in the current manuscript.

Given that this paper also emphasizes the “nice properties” derived from adjusting $\alpha$ within the $\alpha$-divergence family, these prior studies are non-concurrent and directly relevant for establishing the paper’s originality and theoretical positioning. Even considering the domain difference (language model and diffusion model), these references are essential for clarifying what is newly contributed here versus what is conceptually inherited.

(Additional potential weaknesses are further elaborated in the Questions section.)

&nbsp;

**References**

[1] Wu et al., 2024, “α-DPO: Adaptive Reward Margin is What Direct Preference Optimization Needs.” (Preprint)

[2] Gupta et al., 2025, “AlphaPO: Reward Shape Matters for LLM Alignment.” (ICML 2025)

[3] Wang et al., 2024, “Beyond Reverse KL: Generalizing Direct Preference Optimization with Diverse Divergence Constraints.” (ICLR 2024)

[4] Shan et al., 2025, “Forward KL Regularized Preference Optimization for Aligning Diffusion Policies.” (AAAI 2025)

**Questions:**

1. Is it still within the range of “noise robustness” when the downstream evaluation results remain unchanged across 10–40% label flipping?
   * In Tables 2 and 5–8, the evaluation scores exhibit little to no variation as the label-flipping ratio increases. Intuitively, if the flipping ratio rises to 40% (nearly half of the 1M Pick-a-Pic V2 dataset), one would expect a gradual decline in human alignment metrics. For instance, PickScore, which originates directly from the Pick-a-Pic V2 authors, should display a noticeable drop. Moreover, if the proposed $\alpha$-divergence formulation demonstrated the least degradation under increasing noise, that would provide strong empirical evidence that $\alpha$-divergence genuinely mitigates label noise. It is therefore questionable whether maintaining nearly constant evaluation scores across 0–40% flipping can be interpreted as true robustness, since the underlying preference distribution has substantially changed by flipping the labels while the reported outcomes remain unaffected.

2. Typos in Tables 5-8?
   * The explanations and captions in Appendix A.4.2 state that the results correspond to SDXL, whereas the tables themselves label the backbone as SD 1.5. Could these be typographical errors, or do the results indeed correspond to SD 1.5?

---

> ### Author Response · Authors · 2025-11-22
> **Response to Reviewer kScQ (Part 1/3)**
>
> We are grateful for your review and valuable comments, and we hope our response has fully addressed your concerns.
>
> **Q1: Discussion on related work and the conceptual foundation of $\alpha$-DPO**
>
> Thank you for your valuable suggestion. We have added a discussion in the main paper to clarify the differences between our $\alpha$-DPO and related work, and have updated the Related Work section of the manuscript accordingly. Below, we provide a concise conceptual comparison between our approach and other relevant methods.
> - **Comparing with AlphaPO[2] and FKPD[4].**
>     - ***Task Definition:*** We explicitly reformulate the DPO loss as a forward KL–based divergence minimisation problem to address its sensitivity to noisy preference labels. In contrast, AlphaPO [2] applies an $\alpha$-divergence inspired transformation to reshape the reward function, with the primary goal of stabilising preference optimisation rather than modifying the underlying divergence formulation. Meanwhile, FKPD [4] focuses on mitigating out-of-distribution issues by replacing the reverse KL regularisation term with the forward KL regularisation, without reformulating the core DPO objective itself.
>     - ***Formulation:*** Our method treats the DPO Loss as a distribution matching problem under the FKL divergence, which can be formulated as $\mathcal{L_{\text{DPO-Diffusion}}}=\mathbb{E_{\mathbf{x}\sim\mathcal{D}}}\left[\mathbb{D_{\text{KL}}}\left[\bar{p^{\ast}}(\mathbf{x_{0:T}}|\mathbf{c}))||\bar{p_{\theta}}(\mathbf{x_{0:T}}|\mathbf{c})\right]\right]$. In contrast, AlphaPO [2], which is built on $f$-DPO (as we disscussed in the next part), leverages the $\alpha$ transform to reshape the reward function, resulting in $r_{\alpha}(\mathbf{x_{0:T}}|\mathbf{c})=\beta(\frac{1-p_{\theta}(\mathbf{x_{0:T}}|\mathbf{c})^{-\alpha/\mathbf{\mathbf{x_{0:T}}}}}{\alpha})$. Meanwhile, FKPD [4], does not reformulate the DPO optimization target. Instead, it replaces the reverse KL regularization term in Diffusion-DPO with the forward KL regularization, and its learning objective is $\max_{p_{\theta}}\mathbb{E_{\mathbf{c}\sim \mathcal{D}, \mathbf{x_{0}}\sim p_{\theta}(\mathbf{x}|\mathbf{c})}}\left [ r_{\phi}(\mathbf{c}, \mathbf{x})\right ] - \beta\mathbb{D_{KL}}\left (p_{\text{ref}}(\mathbf{x_{0}}|\mathbf{c})||p_{\theta}(\mathbf{x}_{0}|\mathbf{c})\right )$.
>     - ***Conclusion:*** Unlike AlphaPO [2] and FKPD [4], which focus on reward divergence regularisation or KL regularisation for maximum entropy, our method treats DPO as a forward KL–based distribution-matching problem, directly minimising divergence between model and target distributions.
> - **Adoption of $\alpha$ divergence in DPO-style formualtion.**
>     - ***(1)$\alpha$-DPO [1]:*** Unlike our approach, $\alpha$-DPO [1] primarily introduces a hyperparameter $\alpha$ to control the KL regularisation between the optimised policy and the reference policy (*i.e.*, the reward margin of preference pairs), forming a dynamic version of IPO [5]. However, despite this modification, it still fundamentally adheres the original FKL-driven DPO formulation and does not mitigate the intrinsic noise sensitivity caused by the mass-covering behaviour of FKL.
>     - ***(2) AlphaPO [2] and $\mathcal{f}$-DPO [3]:*** Unlike our approach, AlphaPO [2] and $\mathcal{f}$-DPO [3] focus on relpacing the KL divergence regularization term in RLHF objective: $\max_{p_{\theta}}\mathbb{E_{\mathbf{c}\sim \mathcal{D}, \mathbf{x_{0}}\sim p_{\theta}(\mathbf{x}|\mathbf{c})}}\left [ r_{\phi}(\mathbf{c}, \mathbf{x})\right ] - \beta\mathbb{D_{KL}}\left (p_{\theta}(\mathbf{x_{0}}|\mathbf{c})||p_{\text{ref}}(\mathbf{x_{0}}|\mathbf{c})\right )$ with the $\alpha$-divergence, effectively using it as a regularizer between the optimized policy and the reference policy to avoid over-optimization. In contrast, our work leverages the $\alpha$-divergence as the distribution matching regularizer, *i.e.*, aligning the learned preference distribution ($\bar{p_{\theta}}(\mathbf{x_{0:T}}|{\mathbf{c}}) \propto p_{\theta}(\mathbf{x_{0:T}}|\mathbf{c})^{\beta}\cdot p_{\text{ref}}(\mathbf{x_{0:T}}|\mathbf{c})^{1-\beta}$) with the target preference distribution ($\bar{p}^{\ast}(\mathbf{x_{0:T}}|\mathbf{c})\propto p_{\text{ref}}\exp(r(\mathbf{x_{0:T}}, \mathbf{c}))$). This formulation directly addresses the sensitivity to noisy preference labels by leveraging the mode-seeking behaviour of the $\alpha$-divergence.
>
> [1] Wu et al., 2024, “α-DPO: Adaptive Reward Margin is What Direct Preference Optimization Needs.”
>
> [2] Gupta et al., 2025, “AlphaPO: Reward Shape Matters for LLM Alignment.”
>
> [3] Wang et al., 2024, “Beyond Reverse KL: Generalizing Direct Preference Optimization with Diverse Divergence Constraints.”
>
> [4] Shan et al., 2025, “Forward KL Regularized Preference Optimization for Aligning Diffusion Policies.”
>
> [5] Azar et al., 2024, “A general theoretical paradigm to understand learning from human preferences.”

---

> ### Author Response · Authors · 2025-11-22
> **Response to Reviewer kScQ (Part 2/3)**
>
> **Q2: Across the 10%-40% label flipping, only small changes are observed in PickScore values**
>
> Thank you for the valuable comment. We address the concern regarding the magnitude of PickScore changes by first noting that, although its absolute variations are small, PickScore reliably reflects preference alignment, and then explaining that the relatively limited amplitude arises from the low intrinsic sensitivity of PickScore.
>
> - **PickScore's Change Amplitude and Metric Scale.**
>     - ***PickScore Reflects Relative Alignment:*** As shown in Tab. 1 in the main paper, PickScore exhibits a consistent downward trend as the label-flipping rate increases from 10% to 40%. Although the absolute numerical change appears small (*e.g.*, from $22.38$ to $22.00$ for DPO), this is a property of the metric’s scale rather than an indication of robustness. The qualitative degradation in generated images—such as poorer text alignment and reduced visual appeal— is clearly visible in Fig. 1 in the main paper, confirming that PickScore still reliably reflects declines in preference alignment.
>     - ***ImageReward Confirms Metric Reliability:*** In comparison, ImageReward shows a much larger relative drop over the same noise levels (from $0.94$ to $0.73$). Together, both metrics consistently demonstrate that the performance of standard DPO shows clear degradation as the proportion of flipped labels increases.
> - **The Insufficient Amplitude Change in PickScore is a Symptom of its Low Intrinsic Sensitivity.**
>     - ***PickScore’s Limited Responsiveness:*** We argue that the limited change in PickScore amplitude is an intrinsic failure of its scoring mechanism to be sensitive to underlying quality signals, rather than a sign of robust performance.
>     - ***Quantitative Analysis via Coefficient of Variation (CV):*** To quantitatively analyse this issue, we compute the Coefficient of Variation (CV) on normalised PickScore outputs over two subsets of the Pick-a-Pic Test dataset ($2,000$ and $50,000$ randomly selected pairs), where scores were normalised to compute the CV for comparative analysis. The results of the CV score are shown in **R**Table 8. A lower CV indicates a narrower dynamic range and reduced responsiveness to meaningful preference differences.
>     - ***Visual Validation:*** In addition, we include visual comparisons of PickScore and ImageReward outputs in Appendix A.3.5, showing that PickScore remains relatively insensitive even when noticeable degradation in semantic alignment and visual fidelity occurs.
>     - ***Observation and Conclusion:*** From these observations, we conclude the following:
>         - (1) Low CV and Limited Responsiveness (PickScore): PickScore exhibits a low CV (23.12%), revealing that it operates within a narrow value range. This inherently limited variance leads to small observed changes in score amplitude, reflecting its low sensitivity to preference quality.
>         - (2) High CV and High Selectivity (ImageReward): In contrast, ImageReward shows a much higher CV (188.24%), indicating a far more selective scoring mechanism that produces substantially greater score variation across different sample qualities.
>
>         **R**Table 8: Coefficient of Variation of PickScore and ImageReward computed over $2,000$ and $50,000$ randomly selected image pairs.
>         | Metrics | CV(%)-2000 | CV(%)-50000 |
>         | :---: | :---: | :---: |
>         | PickScore | $23.12$ | $22.65$
>         | ImageReward | $\textbf{188.24}$ | $\textbf{189.05}$ |
>
> - **Effectiveness of our $\alpha$-DPO.**
>     - ***Results with 40% Label Flipping:*** Even under 40% label flipping, the majority of preference samples remain correct, providing sufficient signal to guide the learning process. Previous works [1,2] have also used 40% noise as a benchmark to evaluate the robustness of DPO-style models to noisy preference labels. Our experimental results show that while performance naturally degrades as noise increases from 0% to 40%, $\alpha$-DPO consistently preserves preference alignment better than competing approaches, demonstrating strong robustness in high-noise regimes.
>     - ***Overall Effectiveness of $\alpha$-DPO:*** The evaluation of ImageReward further substantiates our conclusions. Across Tabs. 2 and 6-8 in the main paper, $\alpha$-DPO consistently achieves the highest PickScore and ImageReward under all noise settings, confirming that our method more effectively suppresses the negative impact of label noise and yields more reliable preference optimisation than prior baselines.
>
> [1] Chowdhury et al., 2024, “Provably robust dpo: Aligning language models with noisy feedback.”
>
> [2] Fujisawa et al., 2025, “Scalable Valuation of Human Feedback through Provably Robust Model Alignment.”

---

> ### Author Response · Authors · 2025-11-22
> **Response to Reviewer kScQ (Part 3/3)**
>
> **Q3: Typos in Tables 5-8**
>
> Thanks for your reminder. In Tabs. 6–8 in the main paper, we present the results on SDXL. Notably, across all evaluated noise levels, our method consistently outperforms the baseline approaches, further demonstrating the effectiveness and robustness of our $\alpha$-DPO. We have carefully revised the table labels, descriptions, and related expressions throughout the manuscript to ensure clarity and consistency. The main changes are summarised below:
>
> - We have changed the backbone name in Tab.s 6-8 in the main paper from SD1.5 to SDXL.
> - We have corrected the Tab. 11 and Tab. 12 in the main paper to ensure the correct marking of the best and the second value.

---

### Official Review · Reviewer_8b67 · 2025-10-26

**Soundness:** 3
**Presentation:** 2
**Contribution:** 2
**Rating:** 4
**Confidence:** 3

**Summary:**

This paper addresses the sensitivity of diffusion model to noisy labels in T2I. The paper first shows that the DPO objectives are equivalent to minimizing the Forward KL divergence, thus explaining the sensitivity of DPO to noise. The paper then proposed an alpha-DPO method aimed to alleviate the above problem. The alpha in the proposed method can be adjusted in a data-aware way, giving the training process more control over noise tolerance. The method was tested on both synthetic and real world datasets.

**Strengths:**

The main strength is to use alpha-DPO to bypass the over-sensitivity in conventional DPO, as revealed by the FKL analysis. Figure 3 illustrates the difference between the FKL and alpha-divergence in covering noisy distribution, which may account for a small amount of data but cause big penalty in regions with small masses. Addressing this point is a strength of the paper. Results are good.

**Weaknesses:**

A major weakness is that to address the over-sensitivity to noise in conventional DPO, the proposed alpha-divergence incurs hyperparameter such as the mu on line 304 to tune, this increases the complexity of the method.
The essence of the alpha-divergence is a weighted control of the KL measurement, which is good but novelty is not particularly high.
It is not clear why Monte Carlo approximation, starting at line 257, would be a good approximation to p^r and p^g(theta)? Can authors give some reasoning behind this statement.
What is the meaning of Figure 4? Can the authors elaborate?
From line 301 to 303, how is f(x^w, x^l, c) maximized?
For the ablation study of Table 4, it seems there was no meaningful difference in the results for different mu, alpha, beta, and the starting point of dynamic alpha scheduling. Does it mean there was little impact on how to choose these parameters.
Writing can be better as some abbreviations were defined more than once.
While line 250 refers to equation 2, there is no equation 2.

**Questions:**

Please see weaknesses.

---

> ### Author Response · Authors · 2025-11-22
> **Response to Reviewer 8b67 (Part 1/5)**
>
> We are grateful for your review and valuable comments, and we hope our response fully resolves your concerns.
>
> **Q1: Clarification on the Motivation, Necessity, and Novel Contribution of the Proposed $\alpha$-DPO**
>
> - **Contribution of our proposed $\alpha$-DPO.**
>     - ***Motivation:***  As shown in Fig. 3 in the main paper, conventional DPO remains oversensitive to label noise due to the inherently mass-covering behaviour of the forward KL divergence. Even when applying reweighted KL formulations, this tendency persists, leaving the model vulnerable to interference from noisy data. In contrast, the $\alpha$-divergence inherently promotes mode-seeking behaviour, effectively reducing the impact of noisy labels. This fundamental distinction motivates replacing the divergence formulation itself, rather than merely adjusting the weight of the forward KL term.
>     - ***Theoretical Contribution:*** **Importantly, the $\alpha$ divergence is not equivalent to a weighted FKL divergence.** We illustrate the inequality by comparing the generator function $f(u)$ of such two divergences. The generator function of weighted FKL divergence is $f_{FKL}(u) \propto -w\cdot\log{u}$ (w is the reweighting factor), while for $\alpha$ divergence, its generator function is $f_{\alpha}(u) \propto \frac{(u^{1-\alpha} - (1-\alpha)u-\alpha)}{(\alpha{(\alpha-1)})}$ [1]. Unlike ad-hoc FKL reweighting, the $\alpha$ divergence structurally reshapes the nonlinearity of the underlying generator function, fundamentally altering how deviations from the target density are penalised. For example, for two samples $x_{1}$, $x_{2}$ with corresponding $u_{2}=u_{1}^{2}$, here $u_{1}$ and $u_{2}$ are the density of $x_{1}$ and $x_{2}$ respectively. And we have $-w\cdot\log{u_{1}}=\frac{(u_{1}^{1-\alpha} - (1-\alpha)u_{1}-\alpha)}{(\alpha{(\alpha-1)})}$, we can obtain $-w\cdot\log{u_{2}} = -2w\cdot\log{u_{1}}=2\frac{(u_{1}^{1-\alpha} - (1-\alpha)u_{1}-\alpha)}{(\alpha{(\alpha-1)})}$. On the other hand, $\frac{(u_{2}^{1-\alpha} - (1-\alpha)u_{2}-\alpha)}{(\alpha{(\alpha-1)})}=\frac{(u_{1}^{2-2\alpha} - (1-\alpha)u_{1}^{2}-\alpha)}{(\alpha{(\alpha-1)})}$. When $w$ and $\alpha$ are fixed, the equality $2\frac{(u_{1}^{1-\alpha} - (1-\alpha)u_{1}-\alpha)}{(\alpha{(\alpha-1)})}=\frac{(u_{1}^{2-2\alpha} - (1-\alpha)u_{1}^{2}-\alpha)}{(\alpha{(\alpha-1)})}$ does not hold for a wide range of $u_{1}$. This illustrates that $\alpha$-divergence introduces a qualitatively different regularisation.
>     - ***Necessity of $\alpha$-DPO:*** Our proposed $\alpha$-DPO inherits the mode-seeking behaivor of $\alpha$ divergence. Compared with DPO (FKL), it better mitigates the negative impact of noisy preference pairs, improving preference alignment.  Furthermore, the dynamic scheduling mechanism with an adjustable $\alpha$ allows sample-wise divergence regularisation to adapt to varying noise levels, enhancing robustness and stability of training.
> - **Hyperparameter $\mu$.**
>     - (1) Introducing $\mu$ incurs negligible computational overhead (For instance, each iteration takes $1.898$ s with $\alpha$-DPO compared to $1.896$ s for standard DPO).
>     - (2) Across different $\mu$ values, our $\alpha$-DPO consistently outperforms or remains on par with standard DPO, as shown in **R**Table 6 (Tab. 4 in the main paper) (*Response to reviewer 2 Q4*). While selecting an optimal $\mu$ requires minor search, the relative performance gain justifies this effort.
>     - (3) Empirically, we also find that, in most cases, larger $\mu$ values tend to yield better results.
>     - In conclusion, the inclusion of $\mu$ does not pose a practical difficulty but rather contributes positively to the robustness and effectiveness of our method. We have included a discussion of $\mu$'s limitations in the Appendix. A.4.3.
>
> [1] Wang et al., 2024, “Beyond Reverse KL: Generalizing Direct Preference Optimization with Diverse Divergence Constraints.” (ICLR 2024)

---

> ### Author Response · Authors · 2025-11-22
> **Response to Reviewer 8b67 (Part 2/5)**
>
> **Q2: Reasons for using the Monte Carlo to approximate $p^{r}$ and $p^{g(\theta)}$**
>
> - **Necessity of approximating the $p^{r}$ and $p^{g(\theta)}$:** The target distributions $p^{r}$ and $p^{g(\theta)}$ are defined via intractable normalisers (partition functions), making exact computation of expectations infeasible. To address this, we adopt Monte Carlo (MC) estimation, which provides a principled approach to approximate these expectations.
> - **MC Approximation is statistical consistency and unbiasedness:** Under mild conditions, the MC estimator is unbiased and converges almost surely to the true expectation as the number of independent samples increases (Law of Large Numbers)[1]. In our setting, the training dataset used, Pick-a-Pic Training Dataset, contains billions of samples, ensuring that the MC estimates are highly reliable and effectively approximate the true expectations, providing a strong statistical foundation for our method.
> - **Comparing with Numerical integration and Variational Inference:** Numerical integration suffers from the curse of dimensionality, requiring computational complexity that grows exponentially with the space dimension $D$, *i.e.*, $\mathcal{O}(k^{D})$ [2] where $k$ is the number of grid points per dimension. In contrast, Monte-Carlo sampling maintains a convergence rate that is independent of dimensionality. Alternative approximation methods, such as variational inference (VI), produce biased estimators by minimising the KL divergence between a simple proxy distribution $q(x)$ and the true target $p(x)$, which can introduce systematic approximation errors due to the restricted variational family [3].
>
> [1] Robert et al., 1999, “Monte Carlo statistical methods.“
>
> [2] Bellman R., 1966, “Dynamic programming.“
>
> [3] Kingma et al., 2013, “Auto-encoding variational bayes.“

---

> ### Author Response · Authors · 2025-11-22
> **Response to Reviewer 8b67 (Part 3/5)**
>
> **Q3: Explaination for Figure 4 and the $f(\mathbf{x}^{w}, \mathbf{x}^{l}, \mathbf{c})$**
>
> - **Figure 4.**
>     - ***Goal of Fig. 4:*** This figure demonstrates the positive correlation between the value of $f(\mathbf{x}^{w}, \mathbf{x}^{l}, \mathbf{c})$ ($u_{t}(\theta)$ in Eq. 15) and the ΔHPSv2 score.
>     - ***Detail of Fig. 4:*** In the figure, the horizontal axis represents $f(\mathbf{x}^{w}, \mathbf{x}^{l}, \mathbf{c})$ and the vertical axis represents ΔHPSv2. Since ΔHPSv2 reflects whether $(\mathbf{x}^{w}, \mathbf{x}^{l})$ constitutes a correct sample—the larger the ΔHPSv2, the higher the probability of correctness—the observed positive proportional relationship indicates that the $f(\mathbf{x}^{w}, \mathbf{x}^{l}, \mathbf{c})$ can serve as an effective internal metric for estimating sample reliability without relying on external HPSv2 evaluation.
>     - ***Conclusion from Fig. 4:*** This analysis confirms that $f(\mathbf{x}^{w}, \mathbf{x}^{l}, \mathbf{c})$ provides a useful internal signal for identifying reliable preference samples, justifying its use in our dynamic $\alpha$ adjustment mechanism.
> - **$f(\mathbf{x}^{w}, \mathbf{x}^{l}, \mathbf{c})$.** We first explain the introduction of the stop gradient verion of $u_{t}(\theta)$, denoted as $f(\mathbf{x}^{w}, \mathbf{x}^{l}, \mathbf{c})$, and then explain why minimizing $\mathcal{L}_{\alpha\text{-DPO}}$ effectively maximizes the log density of $f(\mathbf{x}^{w}, \mathbf{x}^{l}, \mathbf{c})$.
>     - (1) ***$f(\mathbf{x}^{w}, \mathbf{x}^{l}, \mathbf{c})$ as Auxiliary Indicator for Dynamic $\alpha$ Adjustment:*** We introduce the function $f(\mathbf{x}^{w}, \mathbf{x}^{l}, \mathbf{c})$, **a stop gradient version of $u_{t}(\theta)$**, as an auxiliary indicator for Dynamic $\alpha$ Adjustment. As shown in Fig. 4 in the main paper, the reward margin $f(\mathbf{x}^{w}, \mathbf{x}^{l}, \mathbf{c})$ exhibits a strong positive correlation with the ΔHPSv2 score. This empirical relationship justifies using $f(\mathbf{x}^{w}, \mathbf{x}^{l}, \mathbf{c})$ as a proxy for sample reliability. Consequently, by leveraging this signal, together with the hyperparameter $\mu$, the dynamic adjustment mechanism can increase $\alpha$ for reliable samples and decrease it for noisier ones, enhancing robustness and stability during training. Importantly, $f(\mathbf{x}^{w}, \mathbf{x}^{l}, \mathbf{c})$ is treated strictly as a non-differentiable signal and does not participate in the gradient backpropagation process.
>     - (2) ***Minimizing $\mathcal{L}_{\alpha\text{-DPO}}$ Maximizes the Log Density $f(\mathbf{x}^{w}, \mathbf{x}^{l}, \mathbf{c})$:*** Our loss function is defined as: $\mathcal{L_{\alpha\text{-DPO}}}=\mathbb{E}\left[\frac{1}{\alpha(\alpha-1)} u_{t}(\theta) \left( u_{t}^{\alpha-1}(\theta) - (1-\alpha) u_{t}^{-1}(\theta) - \alpha \right)\right]$. To understand the optimization direction, we analyze the gradient of $\mathcal{L_{\alpha\text{-DPO}}}$ with respect to $u_{t}(\theta)$ ($f(\mathbf{x}^{w}, \mathbf{x}^{l}, \mathbf{c})$). The gradient is calculated as $\nabla_{u_{t}}\mathcal{L_{\alpha\text{-DPO}}}=\frac{1}{(\alpha-1)}({u_{t}^{\alpha-1}}-1)=\frac{1}{(1-\alpha)}(1-\frac{1}{u_{t}^{1-\alpha}})$. Given the constraints $0<\alpha<1$ and $0<u_{t}<1$, we have $\nabla_{u_{t}}\mathcal{L_{\alpha\text{-DPO}}}<0$. This indicates that $\nabla_{u_{t}}\mathcal{L_{\alpha\text{-DPO}}}$ decreases monotonically. Consequently, to minimize the loss $\mathcal{L_{\alpha\text{-DPO}}}$, the model is guided to maximize the $u_{t}(\theta)$ ($f(\mathbf{x}^{w}, \mathbf{x}^{l}, \mathbf{c})$), thereby efectively increasing the likelihood of the preferred sample ($\mathbf{x}^{w}$) over the rejected sample ($\mathbf{x}^{l}$).

---

> ### Author Response · Authors · 2025-11-22
> **Response to Reviewer 8b67 (Part 4/5)**
>
> **Q4: Clarification on the parameter sensitivity observed in Table 4.**
>
> We first clarify the conclusions from the hyperparameter sensitivity ablation study, and then use the Coefficient of Variation (CV) to demonstrate that PickScore exhibits only a small amplitude change across different samples.
>
> - **Clarification of the sensitivity of hyperparameters**
>     - ***PickScore Reflects Relative Alignment:*** Although the absolute variations in PickScore are relatively small, this is a property of the metric’s scale rather than an indication of robustness. Its relative magnitude, however, still consistently reflects whether the generated results are preference-aligned, as also shown in Tab. 1 for SDXL and Fig. 2 in the main paper.
>     - ***ImageReward Confirms Metric Reliability:*** To further validate the analysis, we additionally report ImageReward results. These results, as shown in **R**Table 6, are consistent with those obtained using PickScore, confirming that our evaluation of hyperparameter choices is reliable and well-justified.
>     - ***Conclusion of the sensitivity of hyperparameters:*** Overall, the results indicate that the starting point of the dynamic $\alpha$ scheduling has minimal impact on the model performance, whereas other hyperparameters, such as $\mu$ and $\beta$, can affect the final outcomes depending on their values.
>
>         **R**Table 6: Ablation on hyperparameters $\mu$ and $\beta$ and the Dynamic $\alpha$ Strategy, evaluated on the Pick-a-Pic Test dataset with the fine-tuned SDXL model. (1) **Effect of $\mu$**: As $\mu$ grows, model performance first increases then decreases. (2) **Effect of the Dynamic $\alpha$ Strategy (Fixed-$\alpha$)**: Without dynamic allocation of the $\alpha$, the model performance degrades significantly. (3) **Effect of dynamic $\alpha$ start step**: the increasing of starting step slightly decrease the performance. (4) **Effect of $\beta$**: As $\beta$ increases, model performance first increases and then decreases.
>
>         | $\mu$  | 0.9999 | 0.8  | 0.5  | 0.3  | 0.1  |
>         |:--|:--|:--|:--|:--|:--|
>         | PickScore | 22.51 | 22.45 | 22.42 | 22.36 | 22.31 |
>         | ImageReward | 1.054 | 1.032 | 1.025 | 1.021 | 1.013 |
>
>         | Fixed $\alpha$ | 0.9999 | 0.8  | 0.5  | 0.3  | 0.1  |
>         |:--|:--|:--|:--|:--|:--|
>         | PickScore | 22.45 | 22.41 | 22.42 | 22.31 | 22.29 |
>         | ImageReward | 1.019 | 1.018 | 1.014 | 1.002 | 0.9933 |
>
>         | dynamic $\alpha$ starts | 0 | 50 | 100 | 150 | 200 |
>         |:--|:--|:--|:--|:--|:--|
>         | PickScore | 22.51 | 22.46 | 22.45 | 22.47 | 22.44 |
>         | ImageReward | 1.054 | 1.047 | 1.048 | 1.041 | 1.031 |
>
>         |  $\beta$| 2000 | 3000 | 4000 | 5000 | 6000 |
>         |:--|:--|:--|:--|:--|:--|
>         | PickScore | 22.28 | 22.38 | 22.51 | 22.43 | 22.31 |
>         | ImageReward | 1.022 | 1.031 | 1.054 | 1.042 | 1.002 |
>
> - **The Insufficient Amplitude Change in PickScore is a Symptom of its Low Intrinsic Sensitivity.**
>     - ***PickScore’s Limited Responsiveness:*** We argue that the limited change in PickScore amplitude is an intrinsic failure of its scoring mechanism to be sensitive to underlying quality signals, rather than a sign of robust performance.
>     - ***Quantitative Analysis via Coefficient of Variation (CV):*** To quantitatively analyse this issue, we compute the Coefficient of Variation (CV) on normalised PickScore outputs over two subsets of the Pick-a-Pic Test dataset ($2,000$ and $50,000$ randomly selected pairs), where scores were normalised to compute the CV for comparative analysis. The results of the CV score are shown in **R**Table 7. A lower CV indicates a narrower dynamic range and reduced responsiveness to meaningful preference differences.
>     - ***Visual Validation:*** In addition, we include visual comparisons of PickScore and ImageReward outputs in Appendix A.3.5, showing that PickScore remains relatively insensitive even when noticeable degradation in semantic alignment and visual fidelity occurs.
>     - ***Observation and Conclusion:*** From these observations, we conclude the following:
>         - (1) Low CV and Limited Responsiveness (PickScore): PickScore exhibits a low CV ($23.12\%$), revealing that it operates within a narrow value range. This inherently limited variance leads to small observed changes in score amplitude, reflecting its low sensitivity to preference quality.
>         - (2) High CV and High Selectivity (ImageReward): In contrast, ImageReward shows a much higher CV ($188.24\%$), indicating a far more selective scoring mechanism that produces substantially greater score variation across different sample qualities.
>
>         **R**Table 7: Coefficient of Variation of PickScore and ImageReward computed over $2,000$ and $50,000$ randomly selected image pairs.
>         | Metrics | CV(%)-2000 | CV(%)-50000 |
>         | :--- | :--- | :--- |
>         | PickScore | $23.12$ | $22.65$ |
>         | ImageReward | $\textbf{188.24}$ | $\textbf{189.05}$ |

---

> ### Author Response · Authors · 2025-11-22
> **Response to Reviewer 8b67 (Part 5/5)**
>
> **Q5: Revising the writing**
>
> We thank the reviewer for the valuable feedback. We have carefully revised the entire manuscript, with all modifications clearly highlighted in the updated version. The main changes are summarised below:
>
> - We have ensured that Equation 2 (line 192) is now clearly visible and properly referenced.
> - We have ensured that Equation 3 (line 200) is now clearly visible and properly referenced.
> - We have changed the equation in line 268 from $r(\mathbf{c}, \mathbf{x_{0:T}})$ to $R(\mathbf{c}, \mathbf{x}_{0:T})$ to avoid misunderstanding.
> - We have revised the typos in line 259 to $\mathcal{D_{\alpha}(\mathbf{P}||\mathbf{Q})}= \frac{1}{\alpha(\alpha-1)} \, \mathbb{E}_{x\sim \mathbf{Q}} \Bigg[ \left( \frac{\mathbf{P}(x)}{\mathbf{Q}(x)} \right)^{1-\alpha} - (1 - \alpha) \frac{\mathbf{P}(x)}{\mathbf{Q}(x)} - \alpha \Bigg].$

---

### Official Review · Reviewer_AiHs · 2025-10-31

**Soundness:** 3
**Presentation:** 3
**Contribution:** 3
**Rating:** 6
**Confidence:** 4

**Summary:**

This paper introduces α-DPO, a noise-robust variant of Direct Preference Optimization (DPO) for aligning diffusion models with human preferences. The authors identify the sensitivity of standard Diffusion-DPO to label-flipping noise and propose replacing the Forward KL divergence with α-divergence, along with a dynamic α-scheduling mechanism. The method is evaluated on both synthetic and real-world datasets and shows consistent improvements over existing baselines.

**Strengths:**

1. This work is to systematically address noise robustness in preference alignment for diffusion models, with a solid theoretical grounding in α-divergence.
2. The connection between DPO and FKL divergence is clearly established, and the motivation for using α-divergence is well-justified.
3. Extensive experiments on SD1.5 and SDXL with varying noise levels demonstrate the effectiveness of α-DPO. The method consistently outperforms strong baselines across multiple metrics.

**Weaknesses:**

1. The paper lacks a thorough exploration of α values beyond the dynamic scheduling setup. A fixed-α baseline with a sweep could better contextualize the contribution of the scheduling mechanism.
2. The human evaluation is based on only 20 participants and 40 prompts, which may not be statistically robust.
3. There is a lack of comparison with many baselines.  like [1]SPO(SD1.5 SDXL), [2]SmPO(SD1.5 SDXL),  [3] MaPO(SD1.5 SDXL). These baselines (checkpoints) are open-sourced.
 [1]Aesthetic Post-Training Diffusion Models from Generic Preferences with Step-by-step Preference Optimization. CVPR2025
 [2] Smoothed Preference Optimization via ReNoise Inversion for Aligning Diffusion Models with Varied Human Preferences. ICML 2025
 [3] Margin-aware Preference Optimization for Aligning Diffusion Models without Reference

**Questions:**

1. Have you experimented with fixed α values across a wider range?
2. How does dynamic scheduling compare to the best fixed α?

---

> ### Author Response · Authors · 2025-11-22
> **Response to Reviewer AiHs (Part 1/2)**
>
> We are grateful for your positive review and valuable comments, and we hope our response fully resolves your concerns.
>
> **Q1: Exploration of the fixed $\alpha$ values**
>
> - **Ablation of the fixed-$\alpha$.** In Tab. 4 in the main paper, we evaluate the impact of using a fixed $\alpha$ with values ${0.1, 0.3, 0.5, 0.8, 0.9999}$. The results show two consistent trends: (1) The best fixed-$\alpha$ performance is obtained when $\alpha=0.9999$, achieving a PickScore of $22.45$, which is still lower than the performance of our dynamic scheduling strategy (PickScore $22.51$). (2) cross all $\alpha$ choices, fixed-$\alpha$ settings consistently underperform our dynamic scheduling under the same $\alpha$ values, demonstrating the effectiveness and necessity of adaptively adjusting $\alpha$ based on sample confidence.
>
> - **More fixed-$\alpha$ values.** To further evaluate the behaviour of fixed $\alpha$, we conduct experiments sweeping $\alpha$ from 0 to 1 with a step size of $0.1$. The resulting performance trends are summarised in **R**Table 1.
>
>
>     **R**Table 1: Quantitative experiment results of sweeping $\alpha$ from 0 to 1 with a step size of $0.1$.
>     | Dynamic $\alpha$ - $\mu$ | $0.9999$ | $0.9$ | $0.8$ | $0.7$ | $0.6$ | $0.5$ | $0.4$ | $0.3$ | $0.2$ | $0.1$ |
>     |  :---------------- | :-------- | :----- | :----- | :----- | :----- | :----- | :----- | :----- | :----- | :----- |
>     |    PickScore↑   |     $\mathbf{22.51}$   |  $\mathbf{22.48}$   |  $\mathbf{22.45}$   |  $\mathbf{22.46}$   |  $\mathbf{22.45}$   |  $\mathbf{22.42}$   |  $\mathbf{22.42}$   |  $\mathbf{22.36}$   |  $\mathbf{22.33}$   |  $\mathbf{22.31}$   |
>     | Fixed-$\alpha$ | $0.9999$ | $0.9$ | $0.8$ | $0.7$ | $0.6$ | $0.5$ | $0.4$ | $0.3$ | $0.2$ | $0.1$ |
>     |      PickScore↑      |     $22.45$   |  $22.43$   |  $22.41$   |  $22.43$   |  $22.41$   |  $\mathbf{22.42}$   |  $22.36$   |  $22.31$   |  $22.25$   |  $22.29$   |
>
>
>     The results demonstrate that our dynamic scheduling consistently outperforms the fixed-$\alpha$ settings.
>
> **Q2: Human evaluation**
>
> - **More human evaluation results.** We expanded our human evaluation by including $20$ additional participants. The results of the full evaluation, incorporating all participants, are summarised in **R**Table 2.
>
>     **R**Table 2: Human Evaluation Results. We recruit 40 participants and use 40 diverse prompts to generate the evaluation images.
>     | Human Evaluation | Text Alignment↑ (%) | Visual Attractiveness↑ (%) | Overall Preference↑ (%) |
>     | :---: | :---: | :---: | :---: |
>     | SDXL | $15.9$ | $14.2$ | $17.7$ |
>     | DPO SDXL | $21.1$ | $19.9$ | $20.8$ |
>     | $\alpha$-DPO SDXL | $\mathbf{63.0}$ | $\mathbf{65.9}$ | $\mathbf{61.5}$ |
>
>     Our proposed $\alpha$-DPO method consistently outperforms both SDXL and DPO across key evaluation metrics, achieving higher scores in Text Alignment, Visual Attractiveness, and Overall Preference. These results strongly support the effectiveness of our approach.
>
> - **P-values of human evaluation.** Moreover, we perform pairwise comparisons of the human evaluation results among SDXL, DPO, and $\alpha$-DPO, and compute the corresponding p-values in **R**Table 3.
>
>     **R**Table 3: P-values of the human evaluation results among SDXL, DPO, and $\alpha$-DPO.
>     | SDXL vs DPO | SDXL vs $\alpha$-DPO | DPO vs $\alpha$-DPO |
>     | :---: | :---: | :---: |
>     | p = 1.000e+00 | p = 5.002e-09 | p = 2.961e-09 |
>
>     The small $p$-value ($p < 0.001$) indicates that the observed preference difference is highly unlikely to occur by chance, confirming that $\alpha$-DPO outperforms both DPO and SDXL in human evaluation. This result underscores the statistical robustness of our proposed method.
> - **Prompts of human evaluation.** To ensure a thorough assessment, the human evaluation employed a diverse and comprehensive set of prompts. The complete listing of these prompts is provided in the Appendix. A.3.3.

---

> ### Author Response · Authors · 2025-11-22
> **Response to Reviewer AiHs (Part 2/2)**
>
> **Q3: Comparing with other baselines**
>
> - **Comparing with MaPO.** Although KTO and SPIN are compared on the SD1.5 model in Appendix Tab. 8, we note that MaPO has not released an official SD1.5 checkpoint. To provide a meaningful comparison, we report MaPO’s results on the SDXL model in **R**Table 4. All prompts are sourced from the Pick-a-Pic Test Dataset.
>
>     **R**Table 4: Quantitative comparison with MaPO on SDXL using prompts sourced from the Pick-a-Pic Test Dataset.
>     | Metric | CLIP↑ | HPSv2↑ | PS↑ | IR↑ | Aes↑ |
>     | :---: | :---: | :---: | :---: | :---: | :---: |
>     | MaPO | $0.3251$ | $29.18$ | $22.07$ | $0.853$ | $5.932$ |
>     | Ours | $\mathbf{0.3326}$ | $\mathbf{30.86}$ | $\mathbf{22.51}$ | $\mathbf{1.054}$ | $\mathbf{5.972}$ |
>
> - **Comparing with SPO and SmPO**.
>     - ***SPO and SmPO can be summarised as RM + DPO.*** Both SPO and SmPO are extensions of the core DPO framework, aimed at improving general preference alignment rather than explicitly addressing noise robustness. They rely on an auxiliary Reward Model (RM) to guide DPO training, which we summarise as **RM + DPO**. The effectiveness of both methods is inherently dependent on the quality of the RM: noisy preference data can degrade RM performance and, consequently, compromise preference alignment.
>     - ***Adopting our $\alpha$-DPO to SPO.*** Given this dependency, a direct comparison with our method would be inequitable. Instead, we adopt the RM + DPO structure from SPO/SmPO and replace DPO with our $\alpha$-DPO, forming RM + $\alpha$-DPO. We further evaluate performance on the SDXL model using prompts from the Pick-a-Pic Test Dataset. The results are shown in **R**Table 5. Since SPO and SmPO share the same RM + DPO structure, and considering computation and time constraints, we adopt SPO as the representative baseline when integrating our $\alpha$-DPO. If additional experiments with SmPO combined with our method are necessary, we are happy to include them in the subsequent rebuttal.
>
>         **R**Table 5: Quantitative experimental comparisons between the SPO official checkpoint, our SPO reimplementation, and SPO combined with our $\alpha$-DPO.
>         | Metric | CLIP↑ | HPSv2↑ | PS↑ | IR↑ | Aes↑ |
>         | :---: | :---: | :---: | :---: | :---: | :---: |
>         | SPO | $0.3178$ | $32.27$ | $\mathbf{22.89}$ | $1.107$ | $6.366$ |
>         | SPO* | $0.3119$ | $32.08$ | $22.86$ | $1.116$ | $6.417$ |
>         | SPO+Ours | $\mathbf{0.3201}$ | $\mathbf{32.31}$ | $\mathbf{22.89}$ | $\mathbf{1.134}$ | $\mathbf{6.501}$ |
>
>         Importantly, experimental comparisons between the SPO official checkpoint, our SPO reimplementation, and SPO combined with our $\alpha$-DPO, indicate that our $\alpha$-DPO functions as a complementary component, successfully integrating with other DPO variants and substantially improving their effectiveness.

---

### Author Response · Authors · 2025-12-02
**Summary for the Area Chair**

Dear Area Chair,

We sincerely thank the AC for your careful consideration of our responses. We understand the adjustments to the discussion period and kindly note that, no further feedback was received following the submission of our rebuttal.

Here, we summarize the reviewers' recognized strengths and raised concerns, followed by our point-by-point clarifications. We sincerely thank R1 (AiHs), R2 (8b67), and R3 (kScQ) for their valuable time and insightful feedback.

**Summary of Recognized Strengths**

- ***Writing: clear, well-justified motivation:*** R1 explicitly states that `the motivation is well-justified`, and R2 emphasizes that `use alpha-DPO to bypass the over-sensitivity` is the main strength, R3 gives the positive comment that `the motivation is straightforward`.
- ***Theoretical Contributions: solid theoretical grounding:*** R1 notes that `This paper systematically addresses noise robustness with a solid theoretical grounding` and further highlights that `the connection between DPO and FKL divergence is clearly established`. R2 echoes this view, emphasizing the theoretical insight with `first shows that the DPO objectives are equivalent to minimizing the Forward KL divergence`, and adds that `illustrates the difference between the FKL and alpha-divergence`.
- ***Performance: superior to baselines:*** R1 explicitly praises that our method `shows consistent improvements` and `consistently outperforms strong baselines across metrics`. R2 concurs that `Results are good`. R3 further emphasizes that `α-divergence is superior in nearly every case`, `consistently outperforming previous methods`.
- ***Evaluation: extensive and rigorous:*** R1 highlights the `Extensive experiments on SD1.5 and SDXL with varying noise levels`. and together with R2, they emphasize that `the method is evaluated on both synthetic and real-world datasets`. R3 further remarks that `test on varying axes of datasets, models, and metrics` and that `The empirical scope of the paper is extensive`.

**Summary of Responses**

All revisions are highlighted in the revised PDF.

***Novel Contributions Beyond Related Methods***
- ***Comparing with weighted FKL (R2 W1)***
    * In Appendix A.2.2, we added a theoretical explanation to show that $\alpha$-divergence is not a weighted control of the FKL divergence but a distinct divergence that yields a different geometry.
- ***Comparing with divergence-based methods (R3 W1)***
    * In Sec. 2, we include additional comparison with alpha-DPO, AlphaPO, $f$-DPO, and FKPD. In summary, unlike our method, these approaches: (1) do not establish the connection between DPO and FKL divergence minimization, and (2) modify the KL divergence term in RLHF objective with controlled KL, $\alpha$-divergence, $f$-DPO, or FKL, to regularize the distance between the optimized policy and the reference policy. In contrast, our method uniquely performs distribution matching between the learned preference distribution and the target preference distribution.

***Additional Experiments and Ablations***
- ***Ablation of fixed-$\alpha$ (R1 W1)***
    * Tab. 4 reports the $\alpha$ sweeps (from 0 to 1 with a step size of $0.1$) and includes additional evaluation with ImageReward. Results show dynamic scheduling consistently outperforms fixed-$\alpha$.
- ***Expanded human evaluation and reliability analysis (R1 W2)***
    * We invited 20 new participants and provided prompts, full results, and statistical analysis of $p$-values in Appendix A.3.3. The results remain statistically significant and consistently show that our method outperforms all competing methods.
- ***Comparing with other baselines (R1 W3)***
    * Our Ablation Study (Sec. 4.4) establishes SPO and SmPO can be formulated as Reward Model + DPO. Critically, combining $\alpha$-DPO with these methods successfully confirms its utility as a 'plug-and-play' module for other DPO variants by improving preference alignment.
- ***Insufficient amplitude change in PickScore (R2 W4 / R3 Q1)***
    * A Coefficient of Variation (CV) analysis (Appendix A.3.4) reveals that PickScore is insensitive to subtle preference differences. We integrated ImageReward into our evaluation, whose variation is significantly higher, to validate that $\alpha$-DPO's improvements are robust and not artifacts of metric insensitivity.

***Writing and Presentation Improvements***
- ***Clarification of Figure 4 (R2 W3)***
    * The revised Fig. 4 illustrates that $f(\mathbf{x}^{w}, \mathbf{x}^{l}, \mathbf{c})$ acts as an auxiliary indicator for dynamic $\alpha$ scheduling.
    * Sec. 3.2 includes a theoretical explanation that minimizing $\mathcal{L}_{\alpha\text{-DPO}}$ maximizes the log density $f(\mathbf{x}^{w}, \mathbf{x}^{l}, \mathbf{c})$.
- ***Notational corrections (R2 W5 / R3 Q2)***
    * We corrected and standardized notations in Eqs. 2-3 and throughout the paper.

We hope these revisions have comprehensively addressed reviewers' concerns.

Sincerely,

The Authors

---

### Meta-Review · Area_Chair_a6up · 2026-01-06

**Summary:**

This paper receives two positive initial reviews with scores 6, 6, and one initial negative review with score 4. The major concerns from Reviewer 8b67 with score 4 are motivation and contribution of the proposed method, new hyper parameters are introduced, some technical clarity for the methodology and experiments. Concerns from other reviewers include human evaluation setting, lack of comparison to baselines, and overlooking of several related works on how different divergences articulate the implicit reward under the DPO framework, etc.

**Reviewer Concerns:**

Most concerns seems to be resolved, but the human eval is still conducted on 40 prompts only. The statistical analysis (p-value) is helpful, but some concerns may still remain  (small data set may easily have some bias).

**Reviewer Scores:**

Reviewer 8b67 may increase the score.

---

### Decision · Program_Chairs · 2026-01-26

Accept (Poster)